# Multi-feature clustering of CTCF binding creates robustness for loop extrusion blocking and Topologically Associating Domain boundaries

Li-Hsin Chang [1,6,8], Sourav Ghosh [1,7,8], Andrea Papale [2], Jennifer M. Luppino [3], Mélanie Miranda[1], Vincent Piras [1], Jéril Degrouard[4], Joanne Edouard[1], Mallory Poncelet[1], Nathan Lecouvreur[1], Sébastien Bloyer[1], Amélie Leforestier[4], Eric F. Joyce[3], David Holcman[2,5] & Daan Noordermeer [1] ✉

Topologically Associating Domains (TADs) separate vertebrate genomes into insulated regulatory neighborhoods that focus genome-associated processes. TADs are formed by Cohesin-mediated loop extrusion, with many TAD boundaries consisting of clustered binding sites of the CTCF insulator protein. Here we determine how this clustering of CTCF binding contributes to the blocking of loop extrusion and the insulation between TADs. We identify enrichment of three features of CTCF binding at strong TAD boundaries, consisting of strongly bound and closely spaced CTCF binding peaks, with a further enrichment of DNA-binding motifs within these peaks. Using multi-contact Nano-C analysis in cells with normal and perturbed CTCF binding, we establish that individual CTCF binding sites contribute to the blocking of loop extrusion, but in an incomplete manner. When clustered, individual CTCF binding sites thus create a stepwise insulation between neighboring TADs. Based on these results, we propose a model whereby multiple instances of temporal loop extrusion blocking create strong insulation between TADs.

Mammalian interphase chromosomes adopt a multi-level spatial organization to fit within the compact cell nucleus, while permitting their genomic functions[1,2]. Among the different levels of organization, Topologically Associating Domains (TADs) are visible as insulated domains at the sub-Megabase scale in Hi-C maps[3,4]. Functionally, TADs act as regulatory neighborhoods for gene regulation, DNA replication, recombination and repair[5–8]. TADs are

formed through a process of Cohesin-mediated loop extrusion that creates chromatin loops within the domain, whereas TAD boundaries impede the formation of loops between domains[9,10]. The combination of continuously ongoing and energy-dependent loop extrusion and its subsequent blocking results in the appearance of TADs in population-averaged Hi-C maps—usually averaging chromatin conformation from many thousands of cells—where intra-domain

[1]Université Paris-Saclay, CEA, CNRS, Institute for Integrative Biology of the Cell (I2BC), 91198 Gif-sur-Yvette, France. [2]École Normale Supérieure, IBENS, Université PSL, 75005 Paris, France. [3]Department of Genetics, Penn Epigenetics Institute, Perelman School of Medicine, University of Pennsylvania, Philadelphia, PA, USA. [4]Université Paris-Saclay, CNRS, Laboratoire de Physique des Solides (LPS), 91405 Orsay, France. [5]Churchill College, University of Cambridge, CB3 0DS Cambridge, UK. [6]Present address: MRC Molecular Haematology Unit, MRC Weatherall Institute of Molecular Medicine, Radcliffe Department of Medicine, University of Oxford, and National Institute of Health Research, Blood and Transplant Research Unit in Precision Cellular Therapeutics, OX3 9DS Oxford, UK. [7]Present address: Department of Pathology and Laboratory Medicine, Western University, N6A3K7 London, ON, Canada. [8]These authors contributed equally: Li-Hsin Chang, Sourav Ghosh. ✉e-mail: daan.noordermeer@i2bc.paris-saclay.fr

contacts are generally enriched about twofold over neighboring regions[11,12].

At the large majority of TAD boundaries, binding of the CTCF (CCCTC-binding factor) insulator protein is detected[3,13]. The 15 bp core DNA binding motif within CTCF binding sites (CBSs) has an orientation, with most TADs carrying convergently oriented motifs at their boundaries[5,13–15]. The importance of this orientation has been confirmed by inverting CBSs, which can considerably reduce insulation between TADs[14,16]. How CTCF blocks the loop-extruding Cohesin complex has been dissected through acute depletion of CTCF using Auxin-mediated degrons. In the absence of CTCF binding, loop formation by the Cohesin complex is not perturbed, but the insulation between neighboring TADs is greatly reduced[17,18]. Removal of individual CBSs has mostly been studied relative to their impact on transcriptional regulation. Initial studies, in the context of embryogenesis and carcinogenesis, reported that removal of a CBS permitted the formation of new boundary-spanning enhancer-promoter (E-P) loops that could induce dramatic ectopic activation of certain genes[19,20]. Follow-up studies at other boundaries revealed a more nuanced view, whereby ectopic activation of genes is prevented by clusters of multiple CBSs[21–28].

How perturbations of TAD boundaries, which introduce a twofold insulation between neighboring domains, can have such a dramatic impact on gene activation has been a source of discussion[12,29–31]. Whereas TADs are readily detected in population-averaged Hi-C maps, both single-cell Hi-C and imaging studies indicate that TADs adopt a more heterogeneous or dynamic organization in individual cells, which may include considerable intermingling between neighboring domains[32–35]. Instead of insulated domains, TADs may thus constitute ensembles of actively extruded loops, with boundaries preventing their spread into neighboring domains. Such a model is supported by recent studies that show that loop extrusion is required for the formation of long-range intra-TAD E-P loops[36–38]. TADs may thus represent statistical properties of chromatin, with different cells containing different ensembles of intra-TAD loops but with a shared depletion of loops that cross boundaries[29,31,39]. Indeed, live-cell imaging to determine the frequency of looping between convergent CBSs confirmed they are not permanently in contact (around 5% for a pair of CBSs at a 505 kb distance and 25% for arrays of CBSs at a 150 kb distance)[40,41]. Biophysical modeling of the convergent pair of CBSs at a 505 kb distance furthermore indicated that boundaries are not impermeable, but instead prone to loop extrusion 'readthrough'[40]. If loop extrusion blocking is indeed a leaky process, the clustering of CBSs may be a strategy to improve overall insulation between neighboring TADs. Indeed, the clustering of CBSs has been shown to be a common mechanism to prevent long-range gene activation between TADs[21–23,25].

How clustering of CBSs influences the blocking of Cohesin mediated loop extrusion, and what is the structural impact on the insulation between TADs remains to be determined. We recently showed that most TADs boundaries emerge as extended 'transition zones' where insulation gradually increases from population-averaged Hi-C data[12] (Fig. 1a). These transition zones, generally in the order of 50–100 kb, are roughly similar in size to the regions where CBSs are clustered. Within these zones, DNA from the neighboring TADs is considerable intermingled, which may represent cell-to-cell variation of loop extrusion blocking and readthrough at individual CBSs. Here, we systematically explore how clustering of CBSs distinguishes strong sites of insulation (i.e. TAD boundaries) from weaker sites elsewhere in the mouse genome. Moreover, we developed Nano-C, a multi-contact 3C assay, to identify and characterize extruded chromatin loops. We identify three distinct features of CTCF binding that are enriched at TAD boundaries, consisting of a prevalent clustering of CTCF binding motifs within closely spaced CTCF ChIP-seq peaks with increased peak values. Multi-contact Nano-C subsequently confirms that individual CBSs block loop extrusion, but permit a certain level of read through as

well. Analysis of clustered CBSs, including various perturbations, confirms that individual sites create a stepwise insulation between neighboring TADs. Clustering of CBSs thus provides robustness to loop extrusion blocking and the insulating function of TAD boundaries. Moreover, these results demonstrate an expanded regulatory potential of TAD boundaries, which may help to explain how noncoding structural variation within larger genomic intervals can influence genome-associated processes[19,20,42,43].

## Results

### Multiple features of CTCF binding are clustered at TAD boundaries

To obtain an unbiased inventory of CBSs in mouse embryonic stem cells (mESCs), we performed CTCF ChIP-seq experiments and peak calling without a pre-selected threshold (Fig. 1a, b). The optimal enrichment of CTCF binding motifs within these CBSs was determined using a cut-off based on diminishing returns (Fig. 1b: elbow in the curve and Supplementary Fig. 1a). Subsequently, only CBSs that contain at least one significant motif were retained, resulting in a list of over 83,000 CBSs in the mESC genome (Fig. 1b and Supplementary Data 1). To assess which features of CBSs are enriched around TAD boundaries, we determined insulation scores followed by boundary calling from previously published high-resolution Hi-C data in mESCs[44,45] (Fig. 1a). To analyze different features of CBSs, we stratified each feature into five groups, followed by their distance distribution relative to the nearest TAD boundary (Fig. 1c). Grouping of CBSs based on the distance from the nearest CBS revealed an increasing enrichment within 100 kb from TAD boundaries for all CBSs that are less than 40 kb from another peak (Fig. 1c: top). This enrichment is particularly strong for CBSs that are less than 10 kb from their nearest neighbor within the 50 kb up- or downstream from their TAD boundary. Consequently, over 90% of TAD boundaries contain more than one CBS in the region 50 kb up- or downstream (Supplementary Fig. 1b). Similarly, grouping of CBSs based on their ChIP-seq peak value shows an increasing enrichment within 50 kb from the boundaries (Fig. 1c: middle and Supplementary Fig. 1c). Here, enrichment is particularly strong for the 20% of CBSs with highest peak values within the 50 kb up- or downstream from their TAD boundary.

While determining the optimal significance of CTCF binding motifs, we observed that many CBSs cover multiple CTCF binding motifs (Supplementary Fig. 1d and Supplementary Data 1). Ranking of CBSs either on ChIP-seq peak value or the number of motifs that are covered reveals positive correlations, suggesting that the additional motifs contribute to ChIP enrichment and thus CTCF binding (Fig. 1d and Supplementary Fig. 1e). Among the 83,000 identified CBSs, almost 60% cover more than one binding motif and within those peaks, over 70% of motifs either overlap another motif or are located within 100 bp from their nearest neighbor (Fig. 1e and Supplementary Fig. 1f). To determine if clustering of CTCF motifs within CBSs is a further defining characteristic of TAD boundaries, we determined the enrichment of CBSs relative to their number of motifs. Although CBSs containing any number of motifs are enriched within the 50 kb up- or downstream from TAD boundaries, this enrichment is further increased when more motifs are present (Fig. 1c: bottom). In agreement with the large number of TAD boundaries that contain more than one CBS in the 100 kb window that surrounds them, we find that over 95% of identified TAD boundaries carry more than one motif in the region 50 kb up- or downstream (Supplementary Fig. 1g).

### Multiple CTCF binding motifs can contribute to CTCF binding within the same peak

To determine if multiple binding motifs can contribute to CTCF binding, besides the most significant motif that is by default reported, we used genome editing to remove subsets of motifs within a CBS (Supplementary Fig. 2a, b). CBS 20326 covers 6 significant binding

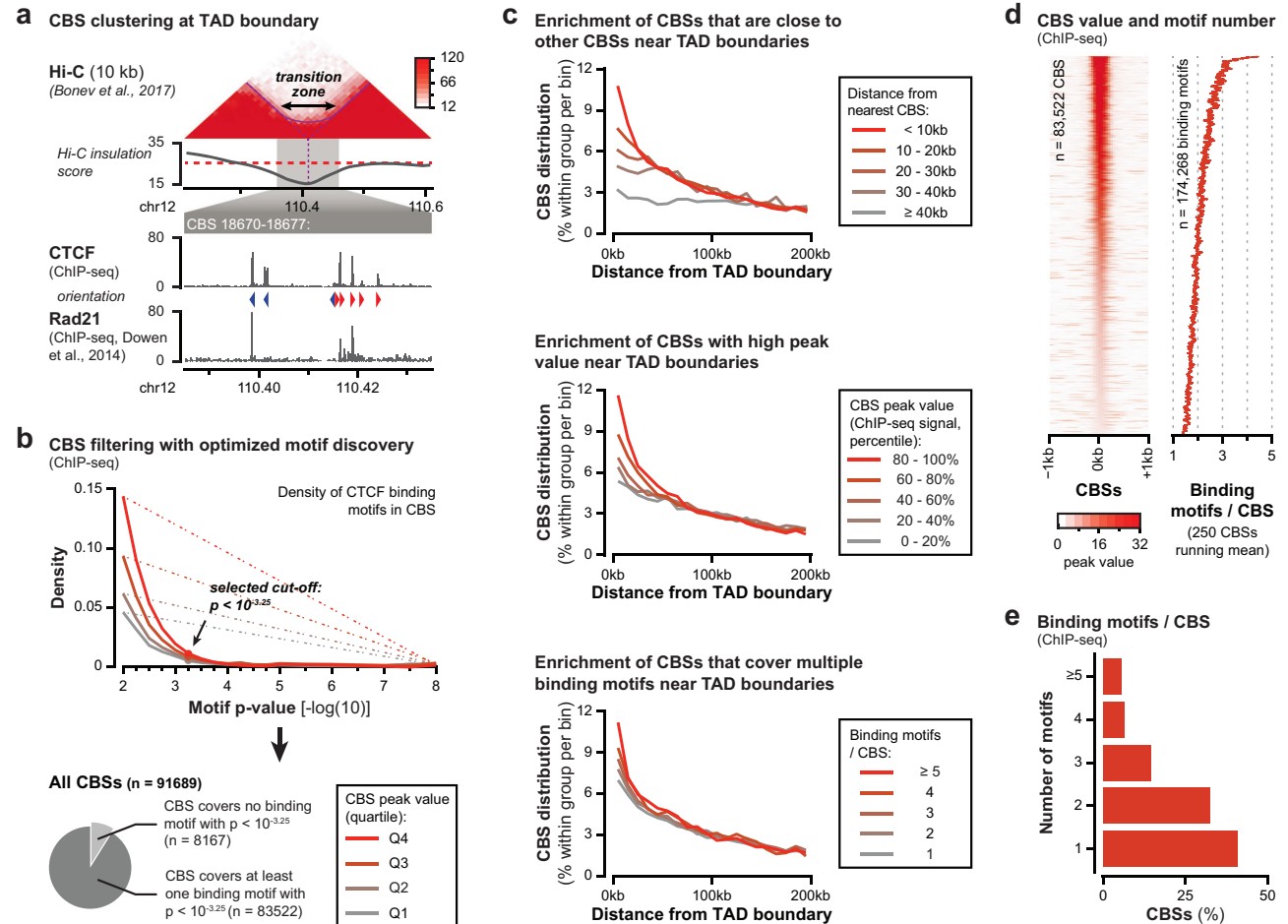

**Fig. 1 | Multiple features of CTCF binding are enriched around TAD boundaries.** **a** Top: TAD boundary in mouse embryonic stem cells (mESCs) that appears as an extended transition zone. Top: intersection of the dashed lines indicates the TAD boundary as called from the minimum in the insulation score, whereas the solid arch demarcates the strong Hi-C signal that indicates intermingling around the boundary. In-between: insulation score (red dashed line: cut-off used for TAD calling). Bottom: zoomed-in CTCF and Rad21 ChIP-seq data showing clustered CBSs within the transition zone. Arrowheads indicate the orientation of the most significant CTCF binding motif in the CBSs. **b** Filtering of CTCF ChIP-seq peaks based on optimal CTCF binding motif discovery. Optimal enrichment is determined based on the maximum distance from a linear increasing density (dotted lines). *P*-values refer to motif significance score as determined using the MEME-suite (see Methods section). **c** Three features of CBSs are enriched close to TAD boundaries in mESCs. Top: CBSs sorted on distance from their nearest CBS relative to TAD boundaries. Middle: CBSs sorted on peak values relative to TAD boundaries. Bottom: CBSs sorted on covered binding motifs relative to TAD boundaries. **d** Ranking of identified CBSs based on peak value (left) and the corresponding number of motifs (running mean). **e** Number of CTCF binding motifs within CBSs.

motifs, of which motif 2 is considered most significant (Fig. 2a, left and Supplementary Data 1). CTCF ChIP-qPCR after removal of motif 2 or motifs 2–6 reveals a strong reduction in CTCF enrichment, yet specific enrichment remains, indicating residual CTCF binding as well (Fig. 2a, right). Conversely, removal of motifs 5–6 reveals a minor reduction in enrichment, despite sites being around 200 bp away from the qPCR target. To determine if perturbed CTCF binding reduced the separation between neighboring TADs, we performed 4C-seq experiments[46] (Supplementary Fig. 2c, d). Using a viewpoint 15 kb upstream from CBS 20326 in WT cells and cell where either motif 2 or the entire CBS 20326 was removed, we detect a moderate increase in contacts that crossed the TAD boundary (see below). This effect is more pronounced when the entire CBS 20326 is removed as compared to motif 2 only (Supplementary Fig. 2c, d). These genome-editing experiments therefore indicate that additional lower significance motifs can contribute to CTCF binding and CBS boundary function.

Next, we wished to validate genome-wide that multiple motifs within CBSs can contribute to CTCF binding. For this purpose, we reanalyzed CTCF SLIM-ChIP data (short-fragment-enriched, low-input, indexed MNase ChIP) in mESCs[47]. Similar to ChIP-exo[48], SLIM-ChIP maps transcription factor-DNA binding at near nucleotide resolution

due to the protection against MNase digestions. Mapping of aggregate SLIM-ChIP signal around all identified CTCF binding motifs reveals a strand-specific peak of signal that covers the 50 bp up- or downstream of the motif center (Fig. 2b, blue and red lines). To assess how the presence of multiple motifs influences binding and protection, CBSs were first separated on containing one or multiple motifs (Fig. 2c and Supplementary Fig. 3a). Next, we categorized motifs on being the most significant, the second most significant or any other rank within their CBS. Moreover, for equal comparison, motifs were further classified on their significance score (Supplementary Data 2). For all categories of motifs, alone or grouped, we observe enrichment and protection, confirming they can all contribute to CTCF binding at the population level (Fig. 2c and Supplementary Fig. 3a).

To obtain a more direct confirmation that multiple motifs within the same CBS can bind CTCF, we analyzed the abundant subset of CBSs where two motifs overlap with a 3 bp shift (Fig. 2d and Supplementary Fig. 3b). Although we consider it highly unlikely that these overlapping motifs are occupied simultaneously, CTCF binding may alternate between them. Consequently, the three unbound bases in the one motif will not be protected against MNase digestion when CTCF binds the other motif. Visually, alternating CTCF binding should thus result

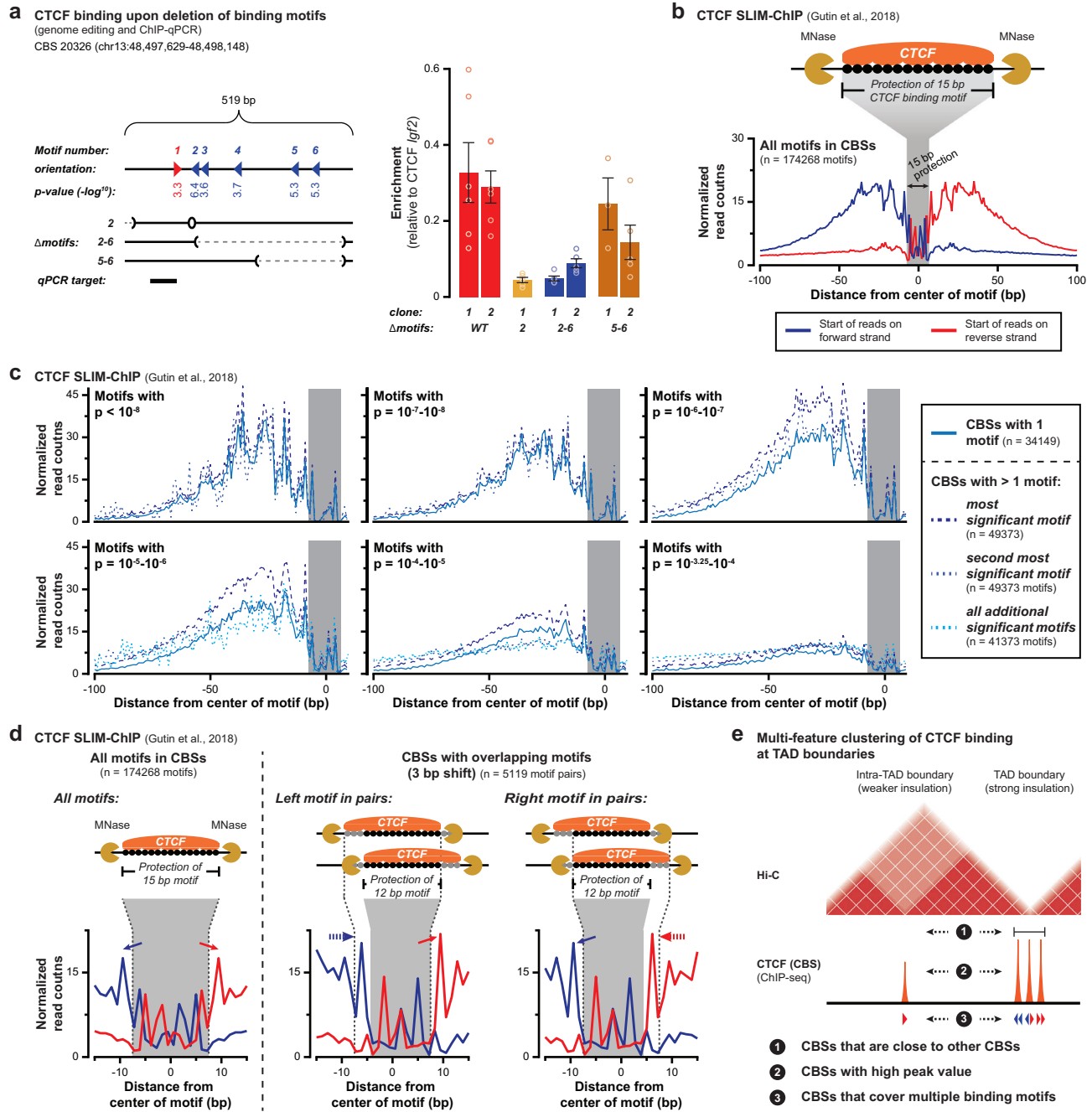

**Fig. 2 | Multiple binding motifs within CTCF binding sites contribute to CTCF binding. a** Removal of CTCF binding motifs within a CBS reveals that multiple motifs contribute to CTCF binding. Left: position of the 6 motifs within CBS 20326, with arrowheads indicating orientation and values indicating significance score. Below, motif deletions and the qPCR target are shown. Right: relative enrichment of CTCF binding as determined by ChIP-qPCR. Data are presented as mean values +/− SEM. Rings indicate individual measurements (*n* = at least 2 measurements each from 2 biological replicates). **b** Outline of the SLIM-ChIP assay to determine nuclease protection at CTCF motifs in WT mESCs. Top: SLIM-ChIP read-out, with black circles indicating protection at the CTCF binding motif. Bottom: normalized pile-up of SLIM-ChIP read start positions around all significant binding motifs in CBSs (200 bp window). The gray area overlaps the 15 bp motif where signal is reduced due to protection by CTCF binding. **c** Normalized pile-up of SLIM-ChIP signal (forward strand only, Supplementary Fig. 3a) for motifs sorted on their

significance (different graphs) and their presence within different groups of CBSs (color coding in legend). Categories with fewer than 100 motifs have been excluded. P-values refer to motif significance score as determined using the MEME-suite (see Methods section). **d** Normalized pile-up of SLIM-ChIP signal for motif pairs that overlap with a 3 bp shift (encompassing ~6% of all motifs). Left: zoomed-in pile-up for all CTCF binding motifs. Middle and right: zoomed-in pile-up of the left and right motifs in overlapping pairs. Solid arrows highlight the two peaks with strong signal that surround the 15 bp binding motif. Dashed arrows highlight the 3 bp shift of the strong peaks, creating a 12 bp protected motif. Above, the impact of alternating CTCF binding on the left or right motif is indicated, which explains the presence of non-protected bases in both motifs, thereby confirming that they can both bind CTCF. **e** Summary of multi-feature enrichment of CTCF binding at strongly insulated TAD boundaries, as compared to sites elsewhere in the genome.

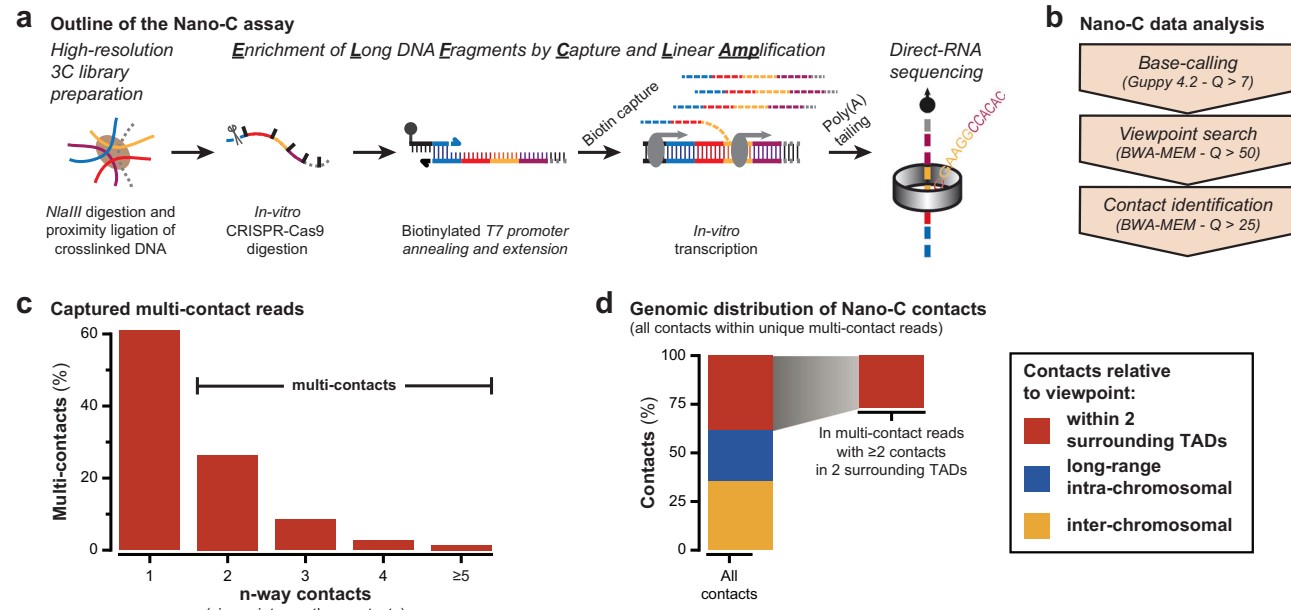

**Fig. 3 | Development of Nano-C technology for multiplexed capture of 3C multi-contacts. a** Overview of the multi-contact Nano-C assay, which combines multiplexed capture of 3C molecules containing pre-selected viewpoints, their linear amplification and their single-molecule direct-RNA sequencing. **b** Stringent three-step filtering of Nano-C reads to identify bona-fide chromatin interactions. **c** Captured pair-wise (1-way) and multi-way (≥2-way) contacts. **d** Genomic distribution of multi-way Nano-C contacts. Further analysis in this study is limited to the 25% of multi-contacts that have at least two contacts in the TADs directly surrounding the viewpoints.

in protection being reduced to 12 bp (Fig. 2d, middle and right). Indeed, both the left and right motifs within the pairs display a different pattern of protection, with strong signal visible within the normally protected binding motif (Fig. 2d and Supplementary Fig. 3c). This reduced protection directly confirms that both motifs within these overlapping pairs can be bound by CTCF.

Based on our analysis of CTCF biding within transition zones, and our validations using genome-editing and SLIM-ChIP analysis, we conclude that TAD boundaries are characterized by an, at least, three-tiered enrichment of CTCF binding (Fig. 2e). At these strong boundaries, CBSs with higher peak values, CBSs that cover more binding motifs and CBSs that are located closer to their neighbors are enriched. The combined action of this multi-feature clustering may provide improved conditions for the blocking of loop extrusion, thereby creating stronger boundaries as compared to weaker intra-TAD boundaries.

## Nano-C can identify 3C multi-contacts for up to 15 multiplexed viewpoints

To determine the loop extrusion blocking capacity of individual CBSs, and thereby their contribution TAD boundary function, we opted for multi-contact 3C to identify higher-order contact hubs with single-allele precision (i.e. formed at a single allele in a single cell). To obtain multiplexed multi-contact information, in a cost-efficient manner, we developed Nano-C (Fig. 3a, b). In a Nano-C experiment, viewpoint-containing 3C concatemers are enriched and amplified by in vitro transcription using a newly developed ELF-Clamp strategy (Enrichment of Long DNA Fragments by Capture and Linear Amplification). First, up to 15 viewpoints of interest are simultaneously digested in vitro using CRISPR-Cas9, followed by the annealing on both extremities of primers containing a biotinylated T7 promoter and DNA polymerase-mediated primer extension. After removal of non-biotinylated fragments, selected fragments are linearly amplified using in vitro transcription. Poly(A)-tailed RNA molecules are subsequently characterized using direct-RNA sequencing on a Nanopore sequencer (Fig. 3a). After stringent filtering of reads, we could typically

map hundreds to thousands of viewpoint-containing multi-contacts in a single experiment (Fig. 3b, c, Supplementary Fig. 4a and Supplementary Data 3). Compared to other targeted multi-contact 3C approaches, i.e. multi-contact 4C (MC-4C) and Tri-C[49,50], Nano-C has the advantage of multiplexing up to 15 viewpoints in a single Nanopore sequencing experiment. This makes Nano-C a cost-efficient choice when elevated numbers of viewpoints are preferred over the number of identified multi-contact reads (Fig. 3c and Supplementary Fig. 4b–d). Comparison of Nano-C results to Hi-C and 4C-seq reveals similar patterns of interactions, confirming that linear amplification and RNA sequencing does not introduce critical biases (Supplementary Fig. 5). In the remainder of this study, we will focus on reads that contain the viewpoint and at least two contacts in the surrounding TADs, which make up around 25% of all viewpoint-containing reads (Fig. 3d).

Like other multi-contact 3C assays that use single-molecule sequencing[49–51], we obtained reads that contained fewer multi-contacts than we expected from the length distributions of 3C libraries (Fig. 3c and Supplementary Figs. 4d and 6a, b). To understand this discrepancy, we used Cryo-EM (Cryo-Electron Microscopy) to visualize a 3C library (Supplementary Fig. 6c). Cryo-EM reveals DNA molecules with lengths of up to 16 kb, with short fragments (<2 kb) being circular or linear. Unexpectedly, longer fragments mostly consist of branched molecules that make up 85% of the DNA content in the 3C library (Supplementary Fig. 6c–e). Proximity ligation of crosslinked DNA thus concatenates multiple single-stranded DNA molecules, with most of these molecules being smaller than the entire 3C concatemer. As long-read sequencing characterizes single-stranded DNA molecules, this explains why the number of identified multi-contacts is reduced.

## Multi-contact Nano-C reveals the higher-order anatomy of extruded DNA loops

To reveal the diversity of multi-contact hubs that are formed by loop extrusion, we use a visualization strategy where each Nano-C read is individually represented (Fig. 4a, see also ref. 52). For each read, the viewpoint is depicted by a black box and its identified multi-contacts

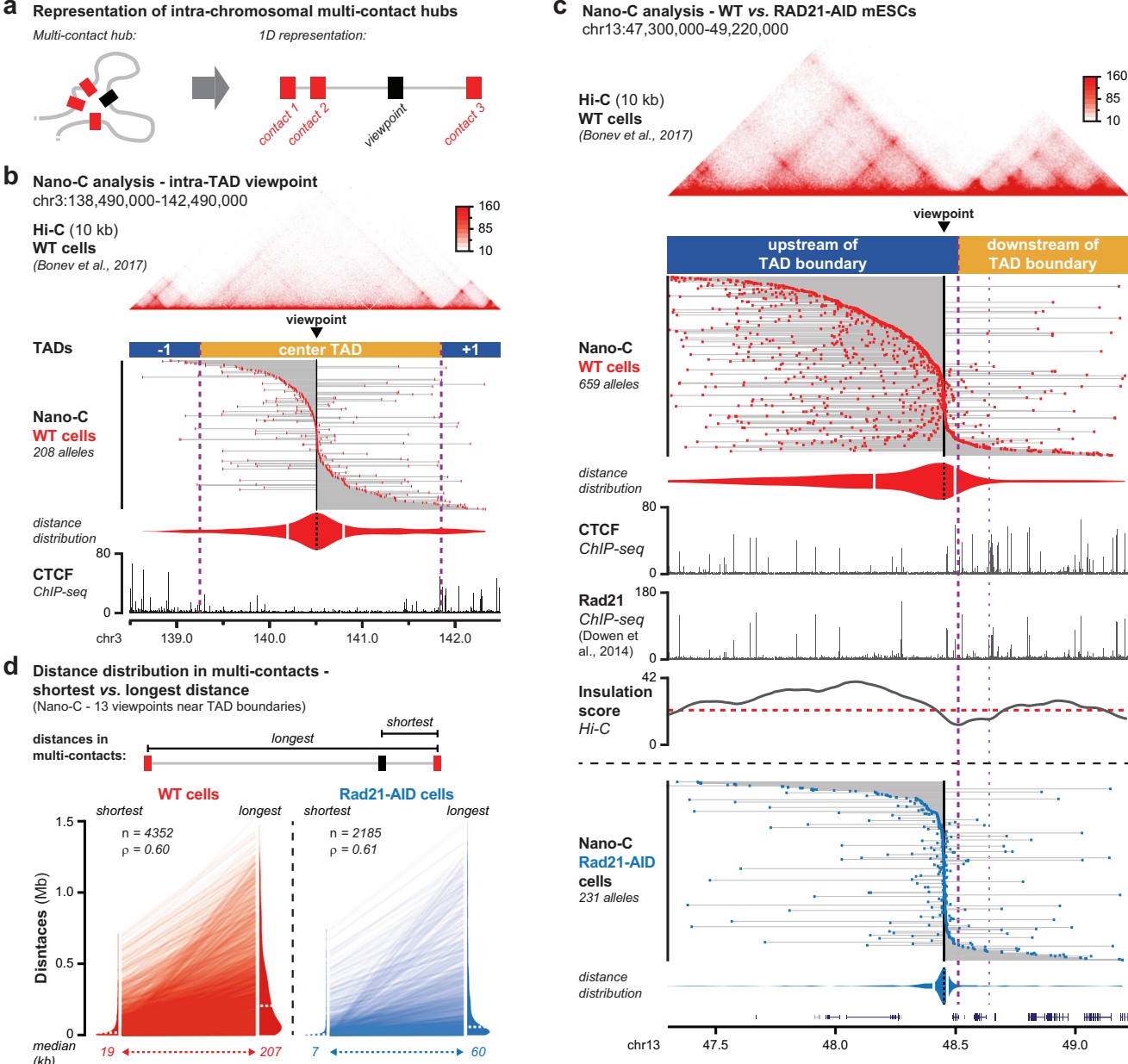

**Fig. 4 | Nano-C identifies and characterizes higher-order loops that structure TADs. a** 1D representation of multi-contacts. The Nano-C viewpoint is indicated with a black box and the identified contacts with red boxes. The gray line connects all contacts, thereby indicating the genomic interval that is spanned. **b** Nano-C multi-contacts in WT mESCs surrounding a central viewpoint within a large TAD with little CTCF binding. Violin plot indicates distances of up- and downstream interactions (white lines: median distances). Dashed purple lines indicate TAD boundaries. The viewpoint and Hi-C data are indicated on top. CTCF ChIP-seq data is indicated below. **c** Nano-C multi-contacts in 2 TADs surrounding a viewpoint that is close to a TAD boundary in WT (top, red) and Rad21-AID (bottom, blue) mESCs.

Violin plots indicate distances of up- and downstream interactions of the viewpoint. The thick purple line indicates the boundary of interest and the thinner line a nearby boundary. Top: WT Hi-C data. In-between: WT CTCF and Rad21 ChIP-seq data, and Hi-C insulation score (red line: cut-off). **d** Shortest and longest distances within Nano-C multi-contacts in WT and Rad21-AID mESCs. Violin plots show the distance distributions for the combined 13 viewpoints (dashed white lines: median distance). Lines between violin plots connect pairs of shortest and longest distances within the same multi-contact. Number of data points and the Spearman correlation score are indicated in the top-left corner.

are depicted by red boxes. Large numbers of reads are subsequently stacked on top of each other to display the diversity of multi-contacts at single alleles within the cell population. First, we determined the diversity of intra-TAD loops by using a viewpoint within a large TAD without prominent CBSs (Fig. 4b). Visualization of over 200 Nano-C multi-contacts, i.e. the viewpoint and at least two interactions, reveals a diverse but mostly symmetrical collection of loops. Whereas an expected decay of interactions over distance is observed (median

distance within three neighboring TADs: ~300 kb), we do not observe any obvious intra-TAD structure (Fig. 4b).

To next assess how loop extrusion influences multi-contact hub formation, we compared Nano-C data from WT and Rad21-AID mESCs, with the latter allowing rapid degradation of the essential Rad21 Cohesin subunit[53] (Supplementary Fig. 7a). Using 13 viewpoints close to TAD boundaries (used for subsequent analyses, see below), major differences in multi-contact distributions are visible (Fig. 4c and

Supplementary Fig. 7b). In WT cells, multi-contacts are spread out over the entire viewpoint-containing TAD. The nearby boundaries impose a rapid drop in interactions though, creating a pattern of asymmetric contacts. In Rad21 depleted cells the number of long-range contacts is drastically reduced, resulting in mostly local and symmetric interactions (Fig. 4c and Supplementary Fig. 7b). Nano-C in WT cells therefore allows to identify and characterize hundreds of DNA loops that incorporate pre-selected viewpoints and that are mostly the result of active loop extrusion.

While we only include multi-contacts in our visualizations (i.e., reads with a viewpoint and at least two separate interactions), we observe many Nano-C reads that give the impression that they link the viewpoint to only one other region in the surrounding TADs (Fig. 4b, c). To better understand this observation, we determined contact distances within Nano-C reads (Fig. 4d and Supplementary Fig. 7c). By focusing on the shortest and longest distance, we find that over 50% of reads include two regions that are less than 20 kb apart. In contrast, particularly in WT cells, the longest span can be any distance within the analyzed TADs. Moreover, this longest span is mostly independent from the shortest distance. Short distances are similar in range in WT and Rad21 depleted cells, suggesting they do not involve active loop extrusion (Fig. 4d and Supplementary Fig. 7c). We therefore envision that multi-contacts that link the viewpoint to a single distant region represent single extruded loops. In contrast, reads with multiple long-range contacts represent higher-order hubs (Supplementary Fig. 7d). The existence of single and higher order hubs was recently inferred from live-cell imaging followed by biophysical modeling[40]. Nano-C provides direct and orthogonal access to such information, by characterizing the higher-order anatomy of extruded DNA loops for up to 15 viewpoints in a single experiment.

### Nano-C confirms the stepwise insulation of clustered CBSs

Next, we used Nano-C to characterize how loop extrusion is blocked by clusters of CBSs within transition zones around TAD boundaries (Fig. 5 and Supplementary Figs. 8, 9). We first focused on a TAD boundary on chromosome 13 where four CBSs are clustered (Fig. 5a, bottom). We designed three viewpoints that either surround the cluster of CBSs or that are localized in the middle. Visual inspection of multi-contacts for each viewpoint reveals a range of loops that spread out over the entire viewpoint-containing TAD. Despite the viewpoints being a short distance from the boundary, loops that cross the boundary are consistently strongly depleted (Fig. 5a and Supplementary Fig. 8). This is confirmed by the distance distributions, which have a differential asymmetry. Whereas intra-TAD contacts of viewpoints 1 and 2 span similar distances, the cross-boundary contacts of viewpoint 2 can reach further into the neighboring TADs (Fig. 5a and Supplementary Fig. 8, violin plots).

To better interpret the higher-order contact hubs that are formed by these viewpoints, we color-coded multi-contact reads (Fig. 5 and Supplementary Figs. 8, 9). Reads where all contacts are left of the boundary are highlighted in blue, reads with all contacts on the right are in orange and reads with contacts on both sides are in gray. Focusing on the TAD boundary on chromosome 13, for all three viewpoints we observe an overrepresentation of multi-contact reads where the viewpoint and all contacts are on the same side of the boundary (Fig. 5a and Supplementary Fig. 8). This observation is confirmed using a statistical approach where enrichment is determined relative to randomized contact distributions (Fig. 5b). Similar enrichments are observed in Nano-C analyses at three other TAD boundaries (Fig. 5c and Supplementary Fig. 9). For all viewpoints, at all TAD boundaries, we further observe that the fraction of reads with the viewpoint and all contacts within the same TAD is stepwise reduced when fewer CBSs separate the viewpoint from the neighboring TAD (Fig. 5b, c). Nano-C thus confirms that individual CBSs contribute to the

overall insulating capacity of extended TAD boundaries, with insulation between domains being reduced in a stepwise manner.

Interestingly, reads where the viewpoint has contacts only on the other side of the boundary are enriched as well (Fig. 5a–c and Supplementary Figs. 8, 9). Although their number is consistently relatively small, their enrichment is much stronger (Fig. 5b, c). These boundary-crossing reads are similarly stepwise increased when fewer CBSs separate the viewpoint from the neighboring TAD (Fig. 5b, c). Conversely, mixed multi-contacts are consistently depleted, independent of the position of the viewpoint relative to the TAD boundary. Combined with our observation that many Nano-C reads constitute single extruded loops (Fig. 4d) and the difference in distance distributions (Fig. 5a, violin plots), we envision two scenarios for the formation of these intra- and inter-TAD loops. In the first scenario, representing intra-TAD loops that obey the boundary, Cohesin loading and loop extrusion are restricted within the same TAD (Fig. 5d, top). In the second scenario, representing inter-TAD loops that cross the boundary, the Cohesin complex is loaded across the TAD boundary from the viewpoint. Loop extrusion read-through of the boundary subsequently creates a loop between the viewpoint and fragments in the other TAD (Fig. 5d, bottom). The enrichment of these boundary-crossing loops thus reinforces the notion that TAD boundaries are not impermeable[40], whereas the stepwise increase of boundary-crossing contacts when fewer CBSs are localized in-between the neighboring TAD confirms that CBSs individually contribute to the blocking of loop extrusion.

### Perturbed CTCF binding increases loop extrusion readthrough of TAD boundaries

To further confirm the stepwise contribution of CBSs to loop extrusion blocking, we performed Nano-C in mESCs where CTCF binding is perturbed. We used cells where either one CBS in a transition zone around a TAD boundary is deleted (ΔCBS 20326 mESCs, Fig. 6a and Supplementary Fig. 2) or in CTCF-AID cells where CTCF is efficiently degraded[17] (Fig. 6a and Supplementary Fig. 10a). ChIP-seq experiments confirm the expected reduction in CTCF binding at either one (ΔCBS 20326 mESCs) or all (CTCF-AID mESCs) CBSs (Fig. 6a). A comparison of the Nano-C distance distributions in WT cells and cells with perturbed CTCF binding reveals no drastic reorganization of interactions (Fig. 6b, c). Instead, more subtle cell type and viewpoint specific changes can be observed. In CTCF-AID cells, all viewpoints form longer-range contacts across the boundary (Fig. 6c). A similar effect is observed in CTCF-AID cells at a different TAD boundary (Supplementary Fig. 10b). Upon depletion of CTCF, the TAD boundary therefore has a reduced capacity to prevent long-range loops between the two neighboring TADs. The deletion of a single CBS has a more subtle effect, with viewpoint 1 moderately extending its contacts within the neighboring TAD as well (Fig. 6c). In contrast, viewpoint 3 does not noticeably change its distance distribution and viewpoint 2, which is interspersed with the CBSs at the boundary, shows a minor increase in long-range contacts within the neighboring TAD (Fig. 6b, c). These differences may be due to orientation or strength of the removed and remaining CBS relative to the viewpoint, thereby permitting an increased influx of loop extruding Cohesin from the downstream TAD (Fig. 6a).

The analysis of multi-contact reorganization upon CBSs perturbations shows a similar pattern (Fig. 6d). In both ΔCBS 20326 and CTCF-AID cells, reads with the viewpoint and all multi-contacts on the same side of the boundary remain overrepresented. However, for viewpoints 1 and 3, their relative abundance is reduced depending if all or only one CBS is affected. Conversely, reads with all interactions on the other side of the boundary are increased as compared to WT cells (Fig. 6d). A similar effect is observed at a different boundary in CTCF-AID cells (Supplementary Fig. 10c). Similar to the distance distribution, perturbation of all CBSs within a transition zone thus increases loop extrusion readthrough of the boundary to a larger extent as compared to the perturbation of a single CBS.

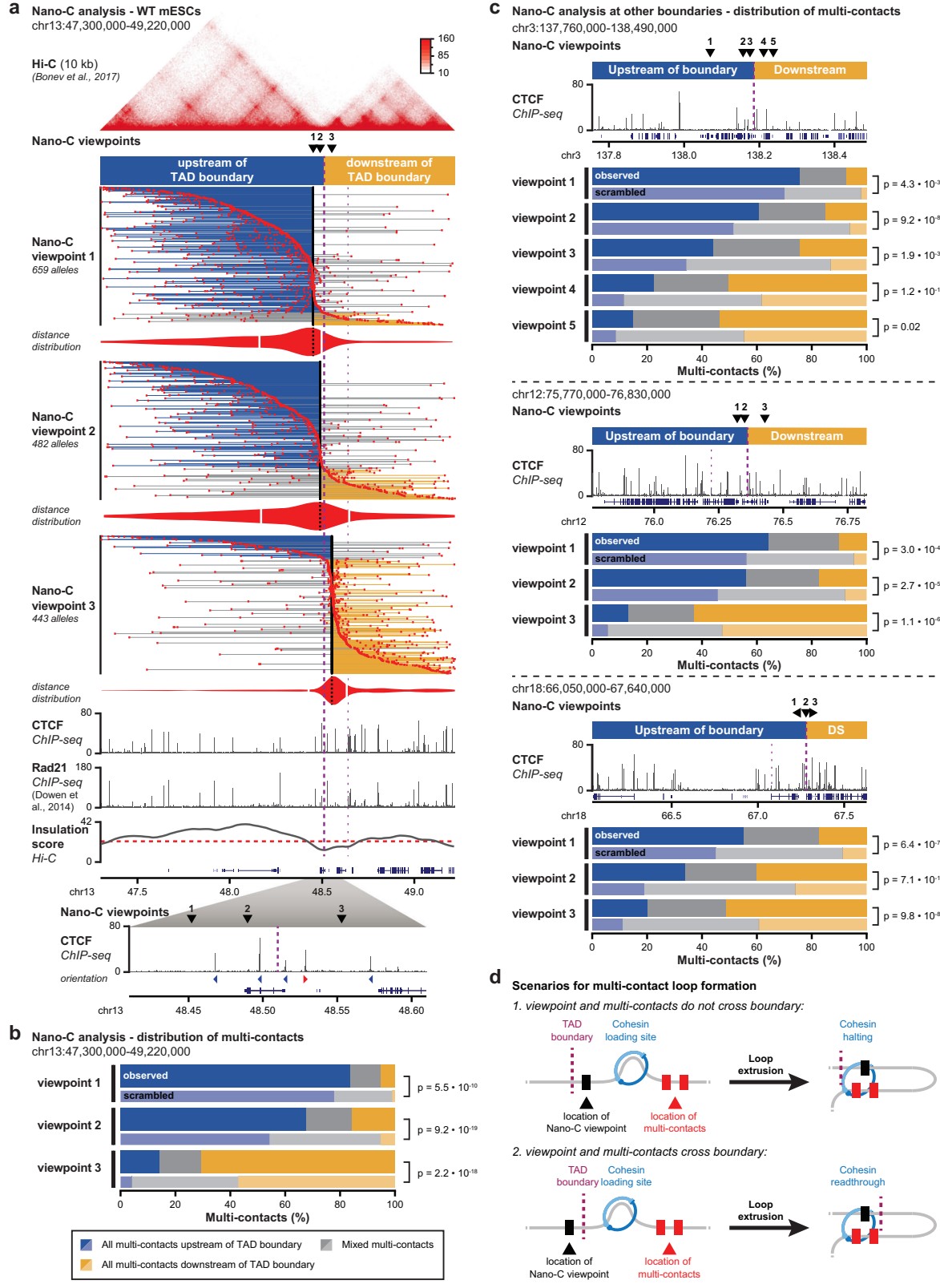

To orthogonally validate the effect of increased Cohesin readthrough upon perturbed CTCF binding, we performed Oligo-paint FISH experiments[34] in the three mESC lines (Fig. 6e). We designed probes in three domains located upstream of the cluster of CBSs or in-between, followed by simultaneous imaging of the domains (Fig. 6e, f and Supplementary Fig. 11a). Compared to WT mESCs, distances between domains are significantly reduced in

cells with perturbed CTCF binding, with the strongest effect in CTCF-AID cells (Fig. 6g and Supplementary Fig. 11b). Depletion of CTCF and the associated increase in loop extrusion readthrough thus caused an observable increase in allelic compaction. Combined, these Nano-C and Oligopaint FISH observations in cells with perturbed CTCF binding confirm that individual CBSs stepwise contribute to loop extrusion blocking and that readthrough of the

**Fig. 5 | Nano-C confirms the stepwise contribution of individual CBSs to loop extrusion blocking and TAD insulation. a** Nano-C multi-contacts for three viewpoints in 2 TADs surrounding a boundary in WT mESCs. The color of the lines that connect multi-contacts indicates if all interactions are upstream of the boundary (blue), downstream of the boundary (orange) or on both sides (gray). Violin plots indicate distances of up- and downstream interactions (white lines: median distances). Viewpoints and Hi-C data are indicated above. ChIP-seq data for CTCF and Rad21, and Hi-C insulation score (red line: cut-off) are depicted below. The thick purple line indicates the boundary of interest and the thinner line a nearby boundary. Below, a zoom-in of CTCF binding in the transition zone surrounding TAD boundary is provided. **b** Distribution of Nano-C multi-contacts in the surrounding TADs. Scrambled distributions of multi-contacts were obtained after randomly assigning contacts up- and downstream into multi-contacts. Significance: G-test of independence. **c** Distribution of Nano-C multi-contacts in the surrounding TADs for viewpoints close to three other boundaries. Above, the position of the viewpoints and CTCF binding is indicated. Significance: G-test of independence. **d** Scenarios of Cohesin loading (left) and extruded loops (right) to explain multi-contacts that do not cross a TAD boundary (top) and that cross a TAD boundary (bottom, involving Cohesin readthrough). Nano-C viewpoints are indicated as black boxes and multi-contacts as red boxes.

boundary is increased in the absence of CTCF binding at one or all CBSs.

## A polymer model incorporating clustered CBSs reproduces transition zones at TAD boundaries

Based on the outcomes of our work, we propose a model whereby clusters of CBSs create strong TAD boundaries by providing a stepwise blocking of loop extrusion (Fig. 7a). In this model, individual CBSs can block loop extrusion but in a non-permanent and possibly incomplete manner. Clustering of CBSs, in combination with a continuous influx of Cohesin, improves the probability or time-period that at least one extruding complex is blocked at a strong boundary, thereby ensuring improved insulation between TADs.

To validate this model, we incorporated different aspects of stepwise and non-permanent loop extrusion blocking into a Randomly Cross-Linked (RCL) polymer model for the simulation of TAD boundaries[54,55] (Supplementary Fig. 12). Specifically, we added the following aspects: fixed connectors at the boundary to model loop extrusion blocking by CTCF; a shifting position of the boundary to model the non-permanent nature of blocking; and an extended boundary to model clusters of CBSs (Fig. 7b). Subsequently, we compared and calibrated this model relative to the average pattern of insulation at all TAD boundaries in mESCs, obtained from reanalyzed Hi-C data (Fig. 7c, dotted black line and Supplementary Fig. 12a). The original RCL model[54,55] uses uniformly distributed random connectors to simulate Cohesin-mediate loop extrusion, with an enrichment of intra-TAD connectors over connectors that cross the boundary is sufficient to create a boundary (Fig. 7c, dashed gray line and Supplementary Fig. 12b). Whereas this model creates an observable TAD boundary, the comparison to experimental Hi-C data reveals notable differences as well. Particularly, in experimental Hi-C, the narrow but deep valley of minimum insulation at the boundary is bordered by two discrete peaks, followed by reduced signal further from the boundary. In contrast, the original RCL model generates a wide valley of minimum insulation with gradually increased signal on either side (Fig. 7c, d, dotted black line versus gray line).

Addition of the three aspects of clustered CBSs each distinctly change the outcome of the RCL model (Fig. 7c, d and Supplementary Fig. 12b). Fixed connectors introduce two discrete peaks of signal, but also create a narrow and shallow valley of insulation. Both a shifting position and an extended size of the boundary widen the valley, but with opposing impact on its depth (Fig. 7c, d and Supplementary Fig. 12b). Combination of fixed connectors with either the shifting or extended boundary improve the model by maintaining the discrete peaks of signal while widening or deepening the valley of minimum insulation. Finally, the inclusion of all three conditions provides and optimal reproduction of the average insulation obtained from Hi-C data (Fig. 7c–e and Supplementary Fig. 12b). The resulting simulated Hi-C matrix has a strong insulation between neighboring TADs, with a zone of enriched contacts emanating from the transition zones that makes up the boundary (essentially an extended 'stripe' that spreads out over a larger region[11]). The addition into the RCL model of these three aspects to model stepwise and non-permanent blocking of loop extrusion thus provides an improved explanation for how clustered CBSs within extended transition zones create insulation between neighboring TADs.

## Discussion

In this study, we determined the structural impact of clustered CBSs on loop extrusion blocking and TAD insulation. Whereas clustering of CTCF binding at TAD boundaries and its impact on the regulation of E-P loops had previously been reported[21–27,56–58], how they create strong insulation between neighboring TADs remained to be addressed. By determining what distinguishes CTCF binding at TAD boundaries from sites elsewhere the genome, we found a multi-featured enrichment of closely spaced CBSs with high ChIP-seq values that cover multiple CTCF binding motifs (Fig. 2e). To determine how such clusters of CBSs, localized in extended transition zones around TAD boundaries[12], influence Cohesin-mediated loop extrusion, we developed and applied multi-contact Nano-C to simultaneously identify and characterize DNA loops for up to 15 pre-selected sites in the genome. First, using cells where the Cohesin complex was depleted, we confirmed that most Nano-C multi-contacts represent actively extruded loops. Next, we showed that individual CBSs have a strong but incomplete capacity to block loop extrusion. Consequently, we detected a stepwise reduction in insulation when fewer CBSs separate a viewpoint from the neighboring TAD. Orthogonal validations, using cells with CTCF perturbations, high-resolution Oligopaint FISH experiments and biophysical simulations further confirmed the dynamic but incomplete nature of loop extrusion blocking by CBSs. Boundary permeability has recently also been inferred from live-cell imaging and biophysical modeling of a pair of pre-selected CBSs[40]. Our Nano-C analysis expands on these observations by identifying the loops that emanate from selected sites in the genome, thereby allowing the direct characterization of loop extrusion blocking by clusters of CBSs.

Instead of constituting insulated domains, TADs have been proposed to represent statistical properties of chromatin[29,31]. In such a representation, TADs emerge from ensembles of extruded intra-TAD loops that are enriched over loops that cross boundaries. Our multi-contact Nano-C analysis confirmed this model by identifying the higher-order loops that include pre-selected viewpoints close to TAD boundaries. The formation of long-range interactions was directly dependent on active loop extrusion, as they were mostly absent in cells where the Rad21 component of the Cohesin complex was depleted. In normal cells, viewpoints formed loops that spread within the entire interval that was covered by their own TAD. In contrast, the presence of a TAD boundary caused a reduction in the both the number and length of the loops that extended into the neighboring domain. By using viewpoints at different positions within extended TAD boundaries, we could directly observe the stepwise contribution of CBSs to the insulation between neighboring TADs. In a similar vein, we could observe that boundary readthrough increases when fewer CBSs were present. These clusters of CBSs roughly overlap the span of the transition zones between TADs in Hi-C maps[12], with the stepwise insulation and permeability of CBSs providing an explanation for the apparent intermingling between the neighboring TADs. Based on these results, we propose a 'traffic light' model for strong TAD boundaries, where individual CBSs block loop extrusion in a temporal manner (Fig. 7a).

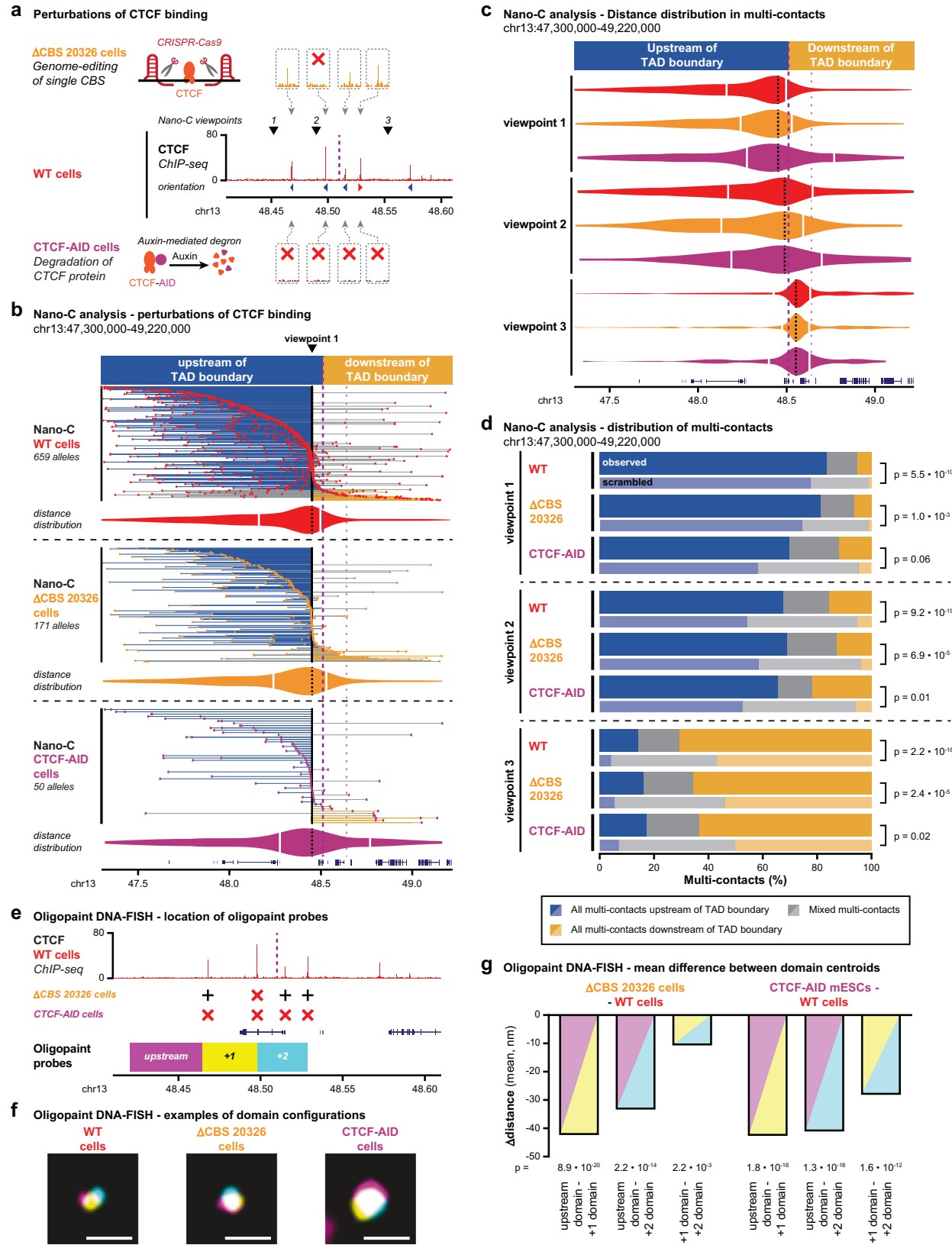

Extended boundaries consisting of clustered CBSs will increase the residence time of the Cohesin complex within the transition zone, which will increase the chance that the complex will dissociate from the chromatin while traversing this region. As a result, readthrough of the boundary will be reduced and these extended transition zones emerge as sites of strong insulation within Hi-C interaction matrix.

Initial studies of CBS orientation reported a strong enrichment of convergently oriented CTCF binding at the extremities of TADs[13–15]. Instead, our analysis of multi-feature CTCF binding enrichment revealed that the large majority of TAD boundaries consist of CBSs with enrichment of three different binding features. Among this enrichment is both a clustering of CBSs (in a range of 50 kb up- and

**Fig. 6 | Impact of CBS perturbations on loop extrusion blocking and TAD insulation. a** Overview of CTCF binding perturbations. Middle: CTCF binding (ChIP-seq) 100 kb up- and downstream of the TAD boundary in WT mESCs. Nano-C viewpoints (arrowheads) and the boundary (purple dashed line) are indicated as well. Above: CTCF binding (ChIP-seq) in the 5 kb surrounding the 4 CBSs directly surrounding the boundary in ΔCBS 20326 mESCs, where a single CBS is removed. Below, CTCF binding (ChIP-seq) surrounding the 4 CBSs in CTCF-AID mESCs, where CTCF is degraded after auxin treatment. Red crosses indicate absence of peaks. **b** Nano-C multi-contacts for a viewpoint in 2 TADs surrounding a boundary in WT, ΔCBS 20326 and CTCF-AID mESCs. The color of the lines indicates if all interactions are upstream of the boundary (blue), downstream of the boundary (orange) or on both sides (gray). Violin plots indicate distances of up- and downstream interactions of the viewpoint (white lines: median distances). The viewpoint is indicated above. The thick purple line indicates the boundary of interest and the thinner line a nearby boundary. **c** Distribution of Nano-C multi-contacts in the surrounding TADs in WT, ΔCBS 20326 and CTCF-AID cells (color-coding as in Fig. 6a). Violin plots indicate distances of up- and downstream interactions of the viewpoint. **d** Distribution of Nano-C multi-contacts in the surrounding TADs in WT, ΔCBS 20326 and CTCF-AID mESCs. Scrambled distributions of multi-contacts were obtained after randomly assigning contacts up- and downstream into multi-contacts. Significance: G-test of independence. **e** Setup of Oligopaint DNA-FISH analysis, with domains covered by probes indicated relative to CTCF binding. Crosses indicate the presence (black cross) or absence (red cross) of CTCF binding in ΔCBS 20326 and CTCF-AID mESCs. **f** Representative Oligopaint DNA-FISH images for the three mESC lines. Scale bar = 1 μm. **g** Difference in mean minimum distance between domain centroids for ΔCBS 20326 and CTCF-AID cells relative to WT cells. Shading refers to the pairs of domains that are analyzed. Significance: two-tailed Mann–Whitney test on pairwise distance distributions.

downstream of TAD boundaries) and of CTCF binding motifs within CBSs (in a range below 1 kb) (Fig. 2e). Grouping of the core CTCF binding motifs within CBSs had previously been reported, but except for a subset of human CBSs, had not been characterized[59–61]. We report that the majority of mouse CBSs cover more than one motif. Interestingly, our reanalysis of CTCF SLIM-ChIP data revealed little difference in binding signal for the most significant binding motifs ($p < 10^{-7}$), independent of their presence in CBSs with one or more motifs. In contrast, intermediate significant motifs ($p = 10^{-4}-10^{-7}$) displayed a considerably higher signal when additional motifs were present in a CBS (Fig. 2c). Grouping of nearby motifs may thus create cooperativity, for instance by promoting favorable nucleosome positioning or by improving the chance of binding after CTCF dissociation[62].

Our analysis of clustered CTCF binding and the stepwise blocking of loop extrusion indicates that most TAD boundaries are of modular nature. The presence of multiple CBSs and motifs increases the fraction of TAD boundaries where loop extrusion can be blocked in both orientations, although the number of motifs in either direction can vary. Combined with protein-protein interactions between CTCF, Cohesin and other proteins at CBSs[63–67], the relative organization of CBSs and motifs at TAD boundaries will have a DNA-encoded regulatory influence. This TAD boundary grammar, consisting of CBSs and their binding motifs within extended regions, will modulate the kinetics of loop extrusion blocking and can help to fine-tune long-range E-P loops. Subsequent site-specific changes to this grammar can influence gene activity, for instance in developmental and evolutionary contexts[15,57,68,69]. The modular nature of TAD boundaries will further buffer against drastic loss of loop extrusion blocking and TAD insulation upon smaller-scale structural variation (e.g. SNPs, local deletions, etc) in disorders and diseases. This may provide a potential explanation for the moderate functional effects that have been observed upon removal or inversion of individual CBSs[14,16,21–23,25]. Conversely, the spreading of TAD boundary function within extended transition zones will enlarge the genomic intervals where structural variation and eQTLs can have an influence on (sub-)TAD insulation and gene regulation[42,43].

## Methods
### Mouse embryonic stem cell culture, CRISPR-Cas9-mediated genome editing and CTCF depletion
Feeder-independent mouse WT mESCs ([male: E14Tg2a.4; a kind gift from Joke van Bemmel and Edith Heard (Institut Curie, Paris, France)[70], mESCs with perturbations in CBS 20326 (this study) and the CTCF-AID and Rad21-AID mESCs [kind gifts from Elphège P. Nora (UCSF, San Francisco, USA), Benoit Bruneau (Gladstone Institutes, San Francisco, USA), Maxim Greenberg (Institut Jacques Monod, Paris, France), Ning Qing Liu and Elzo de Wit (Dutch Cancer Institute, Amsterdam, The Netherlands)[17,53] were cultured on gelatin-coated plates. Cells were grown in DMEM + GlutaMAX (GIBCO, 61965) medium supplemented 0.1 mM NEAA (GIBCO, 11140), 1 mM sodium pyruvate (GIBCO, 11360), 15% FBS (GIBCO, 10270), 10 M β-mercaptoethanol (GIBCO, 21985), 1,000 U/μl of leukemia inhibitory factor (GIBCO, PMC9484). Medium for cells with perturbations in CBS 20326 and the Rad21-AID mESCs was further supplemented with 1 mM MEK inhibitor PD0325901 (Sigma, PZ0162) and 3 mM GSK3 inhibitor CHIR99021 (Sigma, SML1046). Cells were kept in an incubator at 37 °C and 5% $CO_2$.

Perturbations in CBS 20326 were introduced using CRISPR-Cas9-mediated genome editing, by designing gRNAs on both sides of the genomic intervals that were deleted (gRNA sequences in Supplementary Data 4). Sequences were cloned into the pSpCas9(BB)−2A-Puro (PX459) V2.0 plasmid (Addgene #62988; a kind gift from Feng Zhang[71]). 500,000 cells were transfected with 2.5 μg of each plasmid using Lipofectamine 3000 (Thermo Fisher Scientific). Forty-eight hours after transfection, cells were placed under puromycin selection (2 mg/ml) for forty-eight hours. Individual colonies were seeded in 24-well plates, selected using PCR screening (see Genotyping primers in Supplementary Data 4). Deletions were further characterized by cloning PCR products into the pGEM-T system (Promega) and Sanger sequencing of the resulting products from 8–10 individual bacterial clones (Supplementary Fig. 2).

The CTCF protein in the CTCF-AID cells and the Rad21 protein in the Rad21-AID cells were depleted by adding 500 μM of indole-3-acetic acid (IAA, a chemical analog of Auxin; Sigma-Aldrich, I5148) to the medium for 24 h, 48 h, and 96 h, or in the wash-off sample, 48 h followed by 48 h of medium without IAA. Depletion of CTCF and Rad21 was confirmed by standard Western blotting as described[17], using 1:500 dilution of CTCF antibody (Merck-Millipore, 07-729), 1:1000 dilution of Rad21 antibody (Abcam, ab992) and 1:500 dilution of Lamin B1 antibody (Abcam, ab65986). Western blots were scanned on a ChemiDoc imaging system (BioRad) using the Image Lab Touch software (v2.3.0.07) supplied with the machine. Uncropped and unprocessed scans are provided in Supplementary Fig. 13.

### ChIP-seq and ChIP-qPCR
ChIP experiments were performed as previously described[72] with minor modifications. Cells were fixed with 2% formaldehyde solution for 5 min at room temperature, followed by the addition of Glycine to 0.125 M and a PBS wash. Cells were lysed in a buffer containing 50 mM Tris HCl pH 7.5, 30 mM NaCl, 10 mM EDTA, 0.5% NP-40, 1% Triton X-100 and 1× Complete EDTA-free protease inhibitors (Roche, 04693132001) for 10 min incubation on ice, followed by lysis of cell nuclei in a buffer containing 50 mM Tris HCl pH 8.0, 20 mM EDTA, 1% SDS and 1× Complete EDTA-free protease inhibitors for 10 min on ice. Crosslinked chromatin was fragmented to 150–300 bp using a Covaris S220 focused-ultrasonicator device (Covaris). 10 μg of chromatin was diluted in 1 ml ChIP dilution buffer (16.7 mM Tris HCl pH 8.0, 167 mM NaCl, 1.2 mM EDTA, 0.01% SDS, 1.1% Triton X-100, pre-cleared with 80 μl Protein A agarose/Salmon Sperm DNA slurry (Merck-Millipore, 16-157), followed by overnight immunoprecipitation with the following

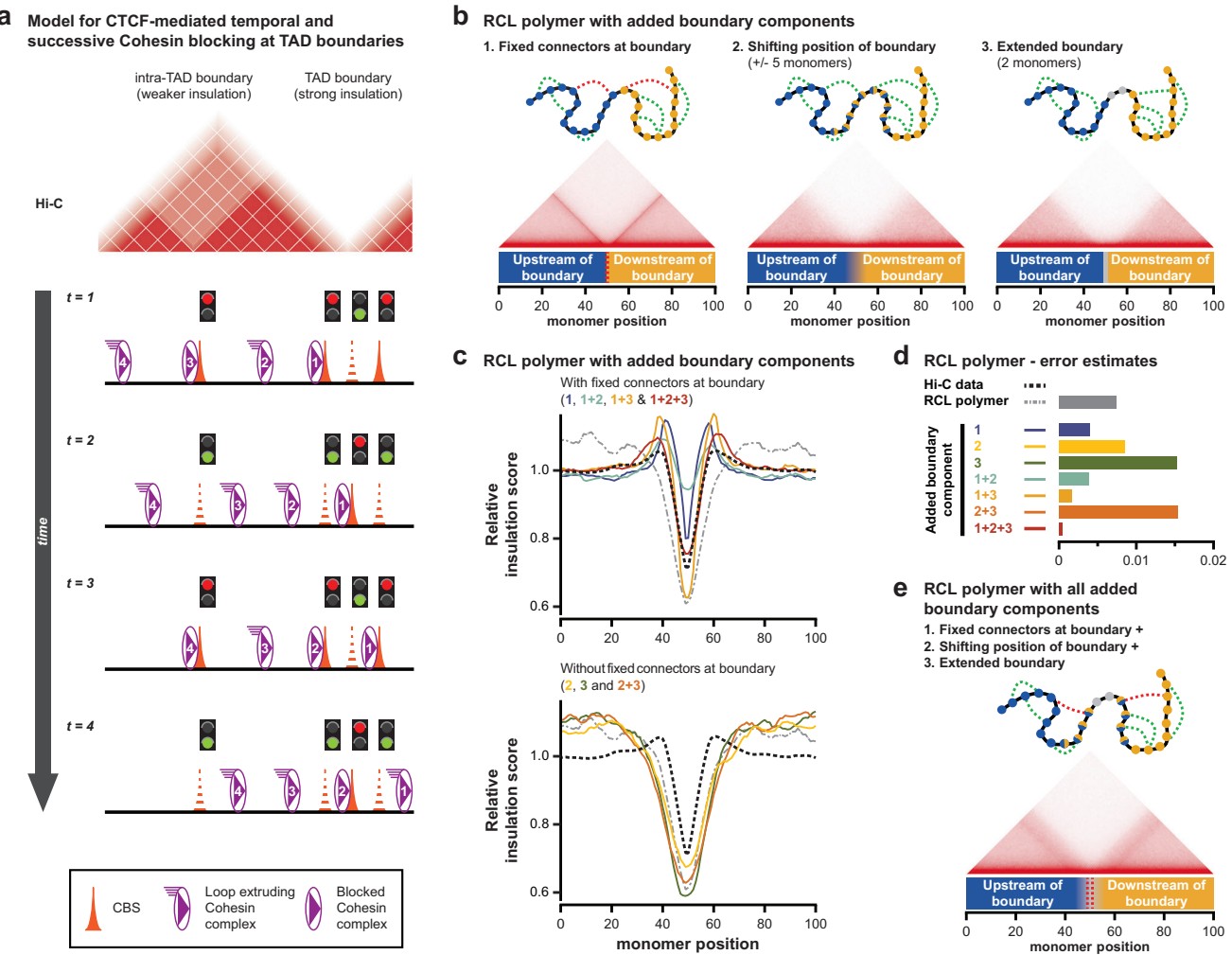

**Fig. 7 | A modified RCL polymer model that incorporates dynamic and clustered CTCF binding improves the simulation of TAD boundary structure and function. a** 1D model for stepwise and non-permanent blocking of Cohesin-mediated loop extrusion by clustered CBSs. Different lines describe the temporal progression of loop extrusion for four Cohesin complexes. Individual CBSs can block extruding complexes, but only in a non-permanent and possibly incomplete manner (exemplified by traffic lights). Isolated CBSs are unable to create long-term blocking of loop extrusion, resulting in weaker insulation between domains. Clustering of multiple CBSs, each inducing temporal blocking, promotes prolonged Cohesin residence at the boundary. Through the continuous influx of Cohesin, this increases the density of the Cohesin complex, thereby creating strong insulation. **b** A modified Randomly Cross-Linked (RCL) polymer model to simulate stepwise and non-permanent loop extrusion blocking at TAD boundaries. Top: scheme depicting the bead-spring chain with added boundary components. Blue monomers belong to TAD 1, orange monomers to TAD 2, blue/orange monomers can

belong to either TAD and gray monomers belong to the gap without connectors. Connectors are placed at random positions within the same TAD (green lines) or fixed at the boundaries (red lines). Bottom: in silico Hi-C map of the 100 monomers that surround the (average) boundary. **c** Relative insulation score for the RCL model with different combinations of added boundary components. Numbers and color coding refer to the added boundary components from Fig. 7b. The black line represents the average insulation score at all boundaries in mESCs, as determined from reanalyzed Hi-C data. The gray line represents the original RCL model without any added boundary components[101]. Top: models with fixed connectors at the boundary; bottom: models without fixed connectors at the boundary. **d** Error estimates for the insulation score in the RCL model with different combinations of added boundary components, relative to experimental Hi-C data from mESCs (smaller values represent a smaller error). **e** An in silico RCL polymer model that incorporates all three aspects of dynamic and clustered CTCF binding results in an improved simulation of TAD boundary structure and function.

antibodies: 5 µg CTCF antibody (Merck-Millipore, 07-729), 2 µl H3K4me3 antibody (Merck-Millipore, 07-473), 5 µg H3K27ac antibody (Active Motif, 39133), 5 µg H3K27me3 antibody (Merck-Millipore, 17-622), 4 µg H3K36me3 antibody (Abcam, ab9050). After addition of 60 µl Protein A agarose/Salmon Sperm DNA slurry, immunoprecipitated chromatin was washed at 4 °C for 5 min using the following buffers: once in Low Salt Immune Complex Wash buffer (20 mM Tris HCl pH 8.0, 150 mM NaCl, 2 mM EDTA, 0.1% SDS and 1% Triton X-100), once in High Salt Immune Complex Wash buffer (20 mM Tris HCl pH 8.0, 500 mM NaCl, 2 mM EDTA, 0.1% SDS and 1% Triton X-100), once in LiCl Immune Complex Wash buffer (10 mM Tris HCl pH 8.0, 250 mM LiCl, 1 mM EDTA and 1% NP-40) and twice in TE (10 mM Tris HCl pH 8.0 and 1 mM EDTA). Chromatin was eluted twice for 15 min at room

temperature in 1 M NaHCO3 and 1% SDS, followed by DNA cleanup and concentration using phenol-chloroform-IAA extraction and ethanol precipitation.

For ChIP-seq, indexed ChIP-seq libraries were constructed using the NEBNext Ultra II Library Prep Kit for Illumina (New England Biolabs, E7645S) and the application note 'Low input ChIP-seq'. Sequencing was done using 50–86 bp single-end reads on the Next-Seq 550 system (Illumina) according to the manufacturer's instructions at the high-throughput sequencing facility of the I2BC (Gif-sur-Yvette, France).

For ChIP-qPCR, enrichment relative to the input of immunoprecipitated chromatin fragments was determined using SsoAdvanced Universal SYBR Green Supermix (Bio-Rad) on a LightCycler480 (Roche) using the software supplied with the machine (v1.5.1.62) or

CFX384 Touch (Bio-Rad) using the built-in software (version number not provided). Enrichment analysis was performed using Microsoft Excel for Mac (v16.75.2). qPCR results are from two PCR experiments each on two biological replicates. Changes in CTCF enrichment at CBS 20326 are expressed relative to a CTCF binding site near the *Igf2* gene on chromosome 7. Primer sequences in Supplementary Data 4.

### ChIP-seq and SLIM-ChIP data analysis

ChIP-seq datasets were mapped to the ENSEMBL Mouse genome assembly GRCm38.p6 (mm10) using BWA (v0.7.15)[73] with default parameters. Rad21 ChIP-seq data from mESCs were obtained from the GEO repository [GSE33346 (https://www.ncbi.nlm.nih.gov/geo/query/acc.cgi?acc=GSE33346)][74].

For ChIP-seq data, duplicate reads, reads with multiple alignments and low-quality reads were removed, followed by the calculation of densities for combined biological and technical replicates. All densities include data from at least two biological replicates, with the exception of the tracks used for visualization in Fig. 6a. For visualization, data densities were further binned in 200 bp windows.

CTCF binding peaks were identified from the combined biological and technical replicates with MACS2 (v2.1.1.20160309)[75] using standard settings and the optimal *q*-value, as estimated on the basis of FDR by the tool itself ($q < 0.05$). De novo discovery of the CTCF consensus binding motif and subsequent assignment of *p*-values to sequence variants was done using the MEME and FIMO tools of the MEME-suite (v5.0.2)[76,77]. We identified all binding motifs with $p \leq 10^{-2}$, allowing multiple motifs per sequence, followed by filtering on the basis of the quartiles of MACS2 peak values and the *p*-value of the binding motifs in steps of 0.25 -log(10) *p*-value. Plotting the number of all identified motifs revealed an obvious elbow in the curve at $p \leq 10^{-3.25}$ for all the quartiles of MACS2 peak values, where the curve bended from a mostly linear to asymptotic increase in identified motifs. CTCF binding peaks were subsequently filtered for the presence of at least one CTCF consensus binding motif with $p \leq 10^{-3.25}$. The list of identified CTCF peaks and their significant CTCF binding motifs is provided in Supplementary Data 1.

CTCF SLIM-ChIP data from mESCs were obtained from the GEO repository [GSE108948 (https://www.ncbi.nlm.nih.gov/geo/query/acc.cgi?acc=GSE108948)][47]. All MNase-digested datasets (1.5 u, 6.25 u, 25 u, 100 u and Mix) were downloaded and mapped as paired-end reads to the ENSEMBL Mouse assembly GRCm38.p6 (mm10) using bowtie2 (v2.4.2)[78] with default parameters, except that maximum fragment length was set to 2 kb. Mapped reads were filtered for mapping quality >20, followed by removal of duplicate reads using the fixmate and markdup options in SAMtools (v1.10, using htslib v1.10.2)[79].

Filtered SLIM-ChIP reads were then used to generate genome wide coverage using the BEDtools suite (v2.30.0)[80], followed by narrow peak calling using MACS2 (v2.2.7.1)[75] with default parameters. Out of the five datasets, MACS2 was unable to build a peak model from the 100 u MNase-digested dataset, which we excluded from our downstream data analysis. The remaining four MNase-digested datasets (1.5 u, 6.25 u, 25 u and Mix) were merged using SAMtools (v1.10), followed by filtering out all the read2 (3'-position within the stranded read) from the paired-end alignment and converting the start points of the read1 alignments into a single base-pair coverage (5'-position within the stranded read). The SLIM-ChIP read distribution in the 100 bp up- and downstream from the midpoint of the CTCF binding motif was next calculated by counting the number of forward strands and reverse strands in each single base-pair window. Further, the read counts were normalized by the total number of CTCF motifs included in each category (Supplementary Data 2).

### Reanalysis of Hi-C data and intersection with ChIP-seq data

Hi-C data from mESCs were obtained from the GEO repository [GSE96107][45]. Reads were mapped to ENSEMBL Mouse assembly GRCm38.p6 (mm10) and processed to aligned reads using HiC-Pro (v2.9.0) and Bowtie2 (v2.3.0), with default settings to remove duplicates, assign reads to DpnII restriction fragments, and filter for valid interactions[78,81]. Hi-C interaction matrices, at 10 kb resolution, were generated from the valid interactions and were normalized using the Iterative Correction and Eigenvector decomposition method (ICE) implemented in HiC-Pro. TAD boundaries were called using TADtool (v0.76)[82], with window size 500 kb and insulation score cut-off value 21.75, resulting in a high degree of genome-wide overlap with TAD borders as reported previously[45]. For the visualization of Hi-C matrices, for the creation of 'virtual 4 C plots' and for the determination of signal densities within TADs, custom R-scripts were used. Intersection of CTCF ChIP-seq peaks with TAD boundaries was done using the BEDtools suite (v2.26.0)[80]. Coordinates of TADs and genome-wide insulation scores are provided in Supplementary Data 5 and 6.

### 4C-seq and data analysis

Chromatin fixation, cell lysis, and 4 C library preparation were done as previously described using 15 million cells per experiment[46]. NlaIII (New England Biolabs, R0125) was used as the primary restriction enzyme and DpnII (New England Biolabs, R0543) as the secondary restriction enzyme. For 4C-seq library preparation, 800 ng of 4 C library was amplified using 16 individual PCR reactions with inverse primers including Illumina TruSeq adapter sequences (primers in Supplementary Data 4). Multiplexed Illumina sequencing was done using 86 bp single-end reads on the Next-Seq 550 system (Illumina) according to the manufacturer's instructions at the high-throughput sequencing core facility of the I2BC (Gif-sur-Yvette, France).

4C-seq data were mapped to ENSEMBL Mouse assembly GRCm38.p6 (mm10), translated to restriction fragments and smoothed (11 fragments running mean) using the c4ctus tool, a stand-alone version of the 4C-seq analysis pipeline that was previously included in the HTSstation tool[83,84]. To determine signal density within TADs or ratios between patterns, the raw values per restriction fragment were used. For visualization, the smoothed values were used.

### Nano-C

High-resolution in-situ Chromosome Conformation Capture (3C) libraries were generated using a conventional 3C protocol[85] with minor modifications. 15 million cells were fixed in a 2% formaldehyde solution for 10 min at room temperature, followed by the addition of Glycine to 0.125 M. Cells were lysed twice by incubation in lysis buffer (10 mM Tris-HCl pH 8.0, 10 mM NaCl, 0.2% NP-40) supplemented with 1× Complete EDTA-free protease inhibitors (Roche, 04693132001) for 20 min on ice. A final concentration of 0.5% SDS was added and extracts were incubated at 62 °C for 10 min. SDS was quenched by addition of Triton X-100 to a final concentration of 1%. Chromatin was then digested with 400 U NlaIII (New England Biolabs, R0125) at 37 °C for 4 h, followed by adding another 400 U at 37 °C overnight. After enzyme inactivation by incubation at 62 °C for 20 min, DNA ligation was performed using 5 μl HC T4 DNA ligase (Promega, M1794) in 1X Ligation Buffer (Promega, C1263) with 1% Triton X-100 and 10 μg/μl BSA and incubation at 16 °C for 4 h. De-crosslinking was performed by adding 50 μl proteinase K (New England Biolabs, P8107) and incubation at 65 °C overnight. 3C libraries were cleaned and concentrated using phenol-chloroform-IAA extraction and ethanol precipitation.

Nano-C experiments were performed using a newly developed ELF-Clamp (Enrichment of Long DNA Fragments using Capture and Linear Amplification) protocol (Fig. 3a), which consists of two selection steps for viewpoints of interest (in vitro CRISPR-Cas9 cutting and site-specific fusion of a biotinylated T7 promoter using primer annealing) followed by specific enrichment using linear amplification (in vitro transcription). The resulting RNA is subsequently characterized using direct-RNA sequencing on an Oxford Nanopore Technologies MinION device[86].

gRNAs for in vitro CRISPR-Cas9 cutting of viewpoints in the 3C libraries were produced by in vitro transcription of a dsDNA containing the gRNA sequence (Supplementary Data 4 for sequences) using the T7 RiboMAX Express Large Scale RNA Production System (Promega, P1320) followed by DNase treatment, according to the manufacturer's instructions. gRNAs were then isolated by 1:1 phenol-chloroform-IAA extraction followed by isopropanol with sodium acetate precipitation.

CRISPR-Cas9 cutting of viewpoints in the 3C libraries was done as follows: 3300 ng each of up to 12 gRNAs was incubated with 33 pmol Alt-R S.p. Cas9 Nuclease V3 (Integrated DNA Technologies) for each gRNA in a total of 25 µl NEBuffer 3.1 (New England Biolabs) at room temperature for 10 min. We have noticed that adding more than 12 gRNAs in a single experiment will result in a reduced number of on-target reads (Supplementary Fig. 4d). 20 µg 3C library was combined with the gRNA-Cas9 complexes and nuclease-free water to a total volume of 250 µl and incubated overnight at 37 °C, followed by enzyme deactivation at 65 °C for 30 min. Nuclear RNA and gRNAs were removed by adding 1250 U RNase $I_f$ (New England Biolabs, M0243) and incubation at 37 °C for 45 min, followed by enzyme deactivation at 70 °C for 20 min. The cut 3C library was purified and concentrated by adding 1 volume AMPure XP beads (Beckman Coulter, A63880) and eluted in 55 µl nuclease-free water. To repair single stranded damage, 53.5 µl of cut 3C library was mixed with 6.5 µl NEBNext FFPE Repair Buffer and 2 µl NEBNext FFPE Repair Mix (New England Biolabs, M6630), followed by incubation at 20 °C for 15 min and addition of 3 volumes of AMPure XP beads for purification.

Site-specific addition of a biotinylated T7 promoter was done by annealing oligos, on both sides of the newly generated cut site, that consisted of the following components: a first biotinylated base, a short linker sequence, the recognition site for the SbfI restriction enzyme (CCTGCA^GG), the complete T7 promoter sequence (TAA-TACGACTCACTATAGGGAG) and a 30 bp sequence that is complementary to the sequence directly bordering the CRISPR-Cas9 cut site (see Fig. 3a and Supplementary Data 4). Oligos were consistently designed for the both sites flanking the cut sites. 0.25 µl each of the 40 µM biotinylated T7 probes on either side of each of the viewpoints were mixed and nuclease-free water was added to the final volume of 10 µl. The biotinylated probe mix was added to 28.5 µl of cut and repaired 3C library, 10 µl of 5X OneTaq buffer, 1 µl of 10 mM dNTPs and 1 µl of OneTaq Polymerase (New England Biolabs, M0480). To generate (partially) double stranded DNA, the reaction was incubated in a thermal cycler with the following steps: 95 °C for 8 min, 1 °C decrease per 15 s to 65 °C, 68 °C for 5 min, rapid decrease to 4 °C. Nuclease-free water was subsequently added to a final volume of 200 µl.

For each probe included in the reaction, 5 µl Dynabeads MyOne Streptavidin C1 beads (Thermo Fisher Scientific, 65001) were combined and washed according to the manufacturer's instructions, followed by resuspension in 200 µl of Binding&Wash buffer (10 mM Tris-HCl, pH 7.0, 1 mM EDTA, 2 M NaCl). The total volume of beads was added to the cut and T7 promoter-fused 3C library, followed by incubation at room temperature for 30 min in a HulaMixer (Thermo Fisher Scientific). Beads were washed three times in 200 µl of 1X Binding&Wash buffer according to the manufacturer's instructions, and further washed in 43 µl nuclease-free water with 5 µl CutSmart Buffer at 37 °C for 20 min. Bound DNA fragments were released by adding 43 µl nuclease-free water, 5 µl CutSmart Buffer, and 20 U SbfI (New England Biolabs, R3642), followed by incubation at 37 °C for 20 min. The released 3C library was purified and concentrated by adding 1 volume AMPure XP beads (Beckman Coulter, A63880) and eluted in 11 µl nuclease-free water.

10 µl of the eluted T7 promoter-fused 3C library was in vitro transcribed using the T7 RiboMAX Express Large Scale RNA Production System (Promega, P1320) according to the manufacturer's instructions with minor modifications: 10 µl of libraries were added to 15 µl 2x T7 RiboMax buffer, 3 µl T7 Enzyme Mix, 2 µl 5 M Betaine

(Sigma-Aldrich, B0300) and 1 µl SUPERaseIn RNase Inhibitor (Thermo Fisher Scientific, AM2694) and incubated at 37 °C for 60 min. Resulting RNA was poly(A) tailed using Poly(A) Tailing Kit (Thermo Fisher Scientific, AM1350) according to the manufacturer's instructions and incubated at 37 °C for 10 min. The poly(A) tailed RNA was purified and concentrated by adding 100 µl of Agencourt RNAClean XP beads (Beckman Coulter, A63987) and eluted in 12 µl nuclease-free water. RNA concentration was determined using the RNA HS Assay on a Qubit device (Thermo Fisher Scientific, Q32852).

Nanopore direct-RNA sequencing libraries were prepared using the Direct-RNA Sequencing Kit (Oxford Nanopore Technologies, version SQK-RNA002) according to the manufacturer's instructions with minor modifications: both adaptor ligation steps were performed for 15 min and the reverse-transcription step was performed at 50 °C for 30 min with 1 µl of SUPERaseIn RNase Inhibitor supplemented. The final Agencourt RNAClean XP beads purification step was done using 24 µl of beads instead of 40 µl for a more stringent size selection. Direct-RNA sequencing was done for 48–72 h, using FLO-MIN106 flowcells (R9 chemistry) on a MinION (MK 2.0) sequencing device with the MinKNOW software (Oxford Nanopore Technologies, latest version available at the time of the experiments).

### Nano-C data analysis

Direct-RNA sequencing reads (fast5 format) were basecalled using Guppy (v4.0.11; Oxford Nanopore Technologies) followed by default quality filtering (QC score > 7), which primarily removes short reads. The resulting RNA fastq files were then converted to DNA fastq using an in-house Perl script.

To reliably identify contacts within our error-prone and complex 3C reads, we devised a two-step mapping and filtering approach. Read quality was evaluated using the MinIONQC tool (v1.4.2)[87]. In the first step, reads in the DNA fastq files were mapped to a synthetic genome only consisting of the different viewpoints present in the run [viewpoint sequences obtained from ENSEMBL Mouse assembly GRCm38.p6 (mm10)] (Supplementary Data 3 for viewpoints contained in each run). We used BWA-MEM (v0.7.15) with default parameters[88] on these complex 3C reads, which considerably outperformed the more commonly used Minimap2 (Nanopore Direct-RNA-seq mode). Retained reads containing the viewpoint were then filtered for high-quality mapping (MQ ≥ 50, which make up the majority of reads; Supplementary Fig. 4a). In the second step, high-quality viewpoint-containing reads were mapped to the entire ENSEMBL Mouse assembly GRCm38.p6 (mm10) with repeats masked (genome obtained from https://www.repeatmasker.org), using BWA-MEM with default parameters. As expected for 3C reads that are composed of fragments from different locations in the genome, we obtained both primary and supplementary mappings from within the same read. Here, we noticed a major difference between the distribution of mapping quality scores for segments that mapped to the same chromosome as the viewpoint (intra-chromosomal mappings; large majority with MQ ≥ 25) and segments that mapped to other chromosomes (inter-chromosomal mappings; large majority with MQ ≤ 25) (Supplementary Fig. 4a). Based on the notion that the large majority of interactions in 3C experiments are intra-chromosomal[89], we assumed that the abundant low quality inter-chromosomal mappings mostly consisted of randomly mapped reads. Individual mapping segments within our viewpoint containing reads were therefore filtered for mapping with quality scores over 25 (MQ ≥ 25).

In-house developed Perl scripts were used to assign the individual segments from each read to NlaIII fragments in the genome, to merge segments falling in the same or neighboring NlaIII fragment and to determine the total number of multi-contacts within the reads. Reads that only mapped to the viewpoint were removed. Numbers of contacts per viewpoint and run are provided in Supplementary Data 3.

Multi-contact information (NlaIII fragments) per read was organized according to the Interact Track Format, with multi-contact reads spanning over multiple lines and additional information provided in non-essential columns:

- *Column 4 (Name): unique identifier assigned to each multi-contact read.*
- *Column 7 (Exp): multi-contacts in the read (viewpoint + n other contacts).*
- *Column 8 (Color): random color assigned to each multi-contact read. #000000 if the originally identified viewpoint was not identified upon mapping to the masked genome.*
- *Column 12 (sourceName): indication if the viewpoint was identified upon mapping to the masked genome (1: yes, 0: no).*
- *Column 17 (targetName): Nano-C runID, indicating the experiment in which the multi-contact read was identified.*

Interact files can be downloaded from the Mendeley Data repository (https://data.mendeley.com/datasets/g7b4z8957z/4). Interact files are based on 3C libraries from at least two biological replicates. Nano-C multi-contact plots were generated from these Interact files after filtering for specific genomic intervals using a custom R-script, with sorting of reads based on the contact that is nearest to the viewpoint.

For benchmarking of Nano-C against Multi-contact 4C (MC-4C) and Tri-C[49,50], the following data sets were obtained from the ENA and GEO repositories:

- *MC-4C: Hbb-b1 viewpoint (ENA study: PRJEB23327; entries: ERR2190825, ERR2190826, ERR2190827, ERR2190831 and ERR2190832).*
- *Tri-C: combined R2, HS-39, HS2 and 3'HS1 viewpoints (GEO study: GSE107940; entries: GSM2878084, GSM2878085, GSM2878087, GSM2878088, GSM2878090, GSM2878091, GSM2878092, GSM2878093, GSM2878094, GSM2878095, GSM2878096, GSM2878097, GSM2878098 and GSM2878099).*

For optimal comparison, MC-4C and Tri-C DNA fastq files were subjected to our newly developed Nano-C data analysis pipeline, with experiment-specific adjustments where needed: for MC-4C, the interactions were translated to DpnII fragments and for Tri-C, the mapping results from both ends of the paired-end reads were combined for further downstream analysis.

### Determination of 3C library topology

To assess if 3C libraries generated from NlaIII-digested chromatin are composed of circular or linear molecules, we treated 200 ng of a 3C library with 0.5 U T5 exonuclease (New England Biolabs, M0663) at 37 °C for 60 min. 200 ng of two control plasmids (9.0 kb and 3.1 kb), with and without linearization, were incubated as well. The presence or absence of degradation was analyzed by gel electrophoresis on a 0.8% agarose gel (Supplementary Fig. 6b).

### Cryo-EM of vitrified 3C libraries

To determine the topology and length of the DNA molecules in a high-resolution 3C library (NlaIII used as restriction enzyme) using cryo-electron microscopy, DNA molecules were trapped suspended within a thin vitreous ice layer and imaged at cryo-temperature[90]. The concentration of the 3C library was adjusted at 80 ng/µl, to ensure that DNA complexes were sufficiently concentrated for imaging but diluted enough not to overlap in most cases. 3 µl of the diluted 3C library was deposited onto a plasma-clean Quantifoil R2/2 holey carbon grid (Electron Microscopy Sciences), blotted with a filter paper for 2 s, and plunged into liquid ethane, using a Vitrobot Mark IV (Thermo Fisher) operated at room temperature and 100% relative humidity. Frozen grids were imaged in a JEOL 2010F transmission electron microscope equipped with a 4 K Gatan Ultrascan 1000 camera at a nominal magnification of ×40,000 or ×50,000. Specimen thickness $t$ was determined as ranging from 50 to 75 nm from stereo pairs recorded at tilt angles ±10°, as described[91]. Images were recorded with a nominal defocus of 3 µm, on the camera (pixel size 0.29 or 0.236 nm) or on Kodak SO163 negative films scanned with a Coolscan 9000 (Nikon) using the Super Coolscan software at a resolution of 4000 pixels per inch. Images were denoised by wavelet filtration in ImageJ ('A trous filter' plugin, with $k_1 = 20$, $k_{n>1} = 0$). DNA contour lengths projected onto the image plane ($L_{//}$) were segmented and measured in ImageJ (v1.53) using the freehand line tool.

The conformation of DNA complexes confined within ice layers of thickness $t$ in the range of the persistence length of the molecule ($l_p ≈ 50$ nm) is described by the Odijk regime of confinement[92,93]: the projected length $L_{//}$ relates to the DNA contour length $L_c$ as $L_{//} ≈ 0.9\,L_c$ (for long molecules $L_c ≫ t$) or, $L_{//} ≈ (π/4)\,L_c$ (for $L_c ≤ t$)[94–96]. Note that with a chain width $w_{DNA} = 2$ nm $≪ t$, the projection of a linear chain on the image plane can cross itself ('self-crossing Odijk regime', $w < t ≤ 2l_p$)[94] and the topology of the complex cannot unambiguously be determined in all cases.

Fifty regions of five specimens frozen from the DNA library were selected for analysis, based upon the presence of well-isolated complexes to exclude the possibility that neighboring molecules overlap. Very large complexes cannot be identified without ambiguities in many cases, leading to a possible underestimation of their number. It is also possible that very small complexes (<10 bp) present in the sample have not been detected. Complexes whose topology could not be unambiguously determined were discarded.

### RNA-seq

RNA-seq experiments were performed as previously described[72] with minor modifications. RNA from 1 million cells was isolated using Trizol LS reagent (Thermo Fisher Scientific, 10296010) followed by rRNA depletion and strand-specific library construction according to the manufacturer's instructions (Illumina). Multiplexed Illumina sequencing was done using 86 bp single-end reads on the Next-Seq 550 system (Illumina) according to the manufacturer's instructions at the high-throughput sequencing core facility of the I2BC (Gif-sur-Yvette, France). Data was mapped to the ENSEMBL Mouse assembly GRCm38.p6 (mm10) using STAR (v2.4.2a) with default parameters[97].

### Oligopaint design and DNA FISH

Oligopaint DNA FISH probes for each domain were designed using the OligoMiner pipeline (upstream domain: chr13:48,420,578–48,464,441; +1 domain: chr13:48,464,441–48,497,888; +2 domain: chr13:48,497,888–48,528,688). Oligopaints were designed to have 80 bases of homology and were purchased from Twist Bioscience. Oligopaints were synthesized as previously described[34] with some modifications to allow for direct conjugation to fluorescent dyes. Specifically, aminoallyl-dUTP (ThermoFisher Scientific) was incorporated into the probes to allow for conjugation with Alexa 488 (ThermoFisher Scientific), Cy3 (Gold Biotechnology), or Alexa 647 (ThermoFisher Scientific).

For FISH experiments, mESCs were fixed in solution with 4% formaldehyde in 1× phosphate-buffered saline (PBS) for 10 min and then washed with PBS. Cells were then settled onto poly-l-lysine coated slides for 30 min at room temperature, fixed again with 4% formaldehyde in PBS for 10 min followed by three 5-min washes in PBS. Slides were stored in PBS at 4 °C. Before FISH, slides were warmed to room temperature (RT) in PBS for 10 min. Cells were permeabilized in 0.5% Triton-PBS for 15 min. Cells were then dehydrated in an ethanol row, consisting of 2-min consecutive incubations in 70%, 90% and 100% ethanol. The slides were then allowed to dry for 2 min at RT. Slides were incubated for 5 min each in 2× SSCT (0.3 M NaCl, 0.03 M sodium citrate and 0.1% Tween 20) and 2× SSCT/50% formamide at RT, followed by a 20 min incubation in 2× SSCT/50% formamide at 60 °C.

 

Hybridization buffer containing primary Oligopaint probes, hybridization mix (10% dextran sulfate, 2× SSCT, 50% formamide and 4% polyvinylsulfonic acid (PVSA)), 5.6 mM dNTPs and 10 µg RNase A was added to slides, covered with a coverslip, and sealed with rubber cement. 2 pmol of probe was used per 25 µl hybridization buffer. Slides were then denatured on a heat block in a water bath set to 80 °C for 30 min, transferred to a humidified chamber and incubated overnight at 37 °C. The following day, the coverslips were removed and slides were washed in 2× SSCT at 60 °C for 15 min, 2× SSCT at RT for 10 min, and 0.2× SSC at RT for 10 min. To stain DNA, slides were washed with Hoechst (1:10,000 in 2× SSC) for 5 min. Slides were then mounted in SlowFade Gold Antifade (Invitrogen, S36936).

Images were acquired on a Leica widefield fluorescence microscope, using a 1.4 NA 63X oil-immersion objective (Leica), an Andor iXon Ultra emCCD camera and the LAS X Software Platform (v3.3, Leica). All images were deconvolved with Huygens Essential (v18.10, Scientific Volume Imaging) using the CMLE algorithm, with a signal to noise ratio of 40 across 40 iterations (DNA FISH) or 2 iterations (DNA stain). The deconvolved images were segmented and measured using a modified version of the TANGO 3D-segmentation plug-in for ImageJ (v0.97)[34,98]. The centroids of all signals were detected using a 3D Gaussian algorithm with a ~20–35 nm $x/y$ resolution and ~40–60 nm resolution in $z$, similar to previous reports[33,99].

### Simulations using the randomly cross-linked polymer model

We describe chromatin organization using our previously developed randomly cross-linked (RCL) polymer model[54]. The RCL polymer has $N_{mon}$ monomers linearly connected by harmonic springs, similar to the Rouse model[100], and an additional $N_c$ random connectors between non-sequential monomers. To represent two sequential TADs, we concatenated two blocks of RCL polymers (each of them composed by $N_{TAD}$ monomers and $N_c$ random connectors)[55,101]. Parameters of the RCL model: $b = 0.18$ µm, $\varepsilon = 0.06$ µm, $D = 8 \times 10^{-3}$ µm²/s (as determined in ref. 55). Numerical simulations are performed with time step $\Delta t = 10^{-2}$ s and $10^6$ integration steps. Hi-C maps are computed by averaging $10^3$ polymer realizations. To investigate which of the identified aspects of CTCF binding at TAD boundaries reproduce best the transition zones between TADs, we simulated different scenarios involving the positioning of random connectors in the RCL block-polymers.

We set $N_{mon} = 2N_{TAD} = 200$ monomers and $N_c = 8$ or 10 random connectors for each RCL polymer; parameters as previously estimated from Hi-C maps[55] and further calibrated using biological Hi-C data (Supplementary Fig. 12a). Subsequently, we investigated different polymer folding scenarios using numerical stochastic simulations. First, we distributed random connectors uniformly within each RCL polymer (Supplementary Fig. 12b, panel "RCL polymer without boundary"). To simulate Cohesin blocking at a punctuated TAD boundary, we next assigned at least one connector in each TAD to join the monomer that is located directly next to the boundary with another monomer randomly chosen within the same TAD (Supplementary Fig. 12b, panel '1. Fixed connectors at the boundary'). To introduce the notion of dynamic CTCF binding, and thus the potential for loop extrusion readthrough, we introduced a moving boundary: the size of the two RCL polymers is randomly chosen $N_{TAD} \in$, keeping the overall size of the polymer constant (panel '2. Shifting position of the boundary'). To optimize the impact of the extended boundary, we set $N_{TAD} = 90$–110 [Supplementary Fig. 12a, shifting position (s)]. We also expanded our model with the addition of an extended boundary (i.e., the presence of a gap containing $N_{GAP}$ monomers without random connectors) that separates the two RCL polymers (panel '3. Extended boundary'). To optimize the extent of the shifting boundary, we set $N_{TAD} = 97$–100 and $N_{GAP} = 0$–5 [Supplementary Fig. 12a, extended boundary (e)]. Next, we combined different combinations of variables: we paired the fixed connectors at the boundaries with the shifting position of the boundary (panel '1. + 2.'), we mixed the fixed connectors with the extended boundary (panel 1. + 3.) and we matched the shifting position of the boundary with the extended boundary (panel '2. + 3.'). The final scenario includes all variables: the fixed connectors at the boundary, the shifting position of the boundary and the extended boundary (panel '1. + 2. + 3.').

### Reporting summary

Further information on research design is available in the Nature Portfolio Reporting Summary linked to this article.

## Data availability

The data that support this study are available from the corresponding author upon request. The unprocessed Oxford Nanopore Technologies and Illumina sequencing data generated in this study (Nano-C, ChIP-seq, 4C-seq, RNA-seq) have been deposited in the European Nucleotide Archive (EMBL-EBI ENA) database under accession code PRJEB44135. The processed sequencing data (Nano-C, ChIP-seq, 4C-seq, RNA-seq) and Cryo-EM images are available at the Mendeley Data repository under accession code 10.17632/g7b4z8957z.4. Unprocessed and processed data are available without restrictions.

## Code availability

Custom code for the analysis of Nano-C data is available from https://github.com/NoordermeerLab/Nano-C. The c4ctus tool for 4C-seq analysis[84] is available from https://github.com/NoordermeerLab/c4ctus. Code is available without restrictions. All other analyses were done using commonly available tools, as described in the Methods section.

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

## Acknowledgements

We thank Alessandra Montecucco (IGM-CNR, Pavia, Italy) and members of the Noordermeer lab for useful discussion. We thank Joke van Bemmel, Edith Heard, Elphège P. Nora, Benoit Bruneau, Maxim Greenberg, Ning Qing Liu, and Elzo de Wit for sharing of mESC lines. We thank Benoit Moindrot (I2BC, Gif-sur-Yvette, France) for help with Western blotting. This work has benefited from the facilities and expertise of the high-throughput sequencing core facility of the I2BC (Gif-sur-Yvette, France). D.N. and E.F.J. benefit from collaborative funding from the Agence Nationale pour la Recherche (ANR-21-CE12-0034-01) / National Science Foundation (NSF). D.N. and D.H. benefit from collaborative funding from PlanCancer (19CS145-00). D.N. was further supported by the Agence Nationale pour la Recherche (ANR-18-CE12-0022-02, ANR-17-CE12-0001-02, ANR-16-TERC-0027-01, ANR-14-ACHN-0009-01), the Fondation Bettencourt Schueller, the Fondation de Coopération Scientifique Campus Paris-Saclay (2015-0980i) and Oxford Nanopore Technologies (MinION Early Access Program and Direct-RNA Early Access Program). S.G. and A.P. were supported by postdoctoral fellowships from the Fondation pour la Recherche Médicale (Post-doctorat en France -

SPF201909009328 and SPF201909009284). Cryo-EM experiments were supported by the CNRS network Microscopie Electronique et Sonde Atomique (METSA, FR CNRS 3507) to D.N. and the Investissements d'Avenir LabEx PALM (ANR-10-LABX-0039-PALM) to A.L. J.M.L. was supported by the National Institute of Health (F31HD102084). E.F.J. was supported by the National Institute of Health (R35GM128903 and U01DA052715). D.H. was supported by the European Research Council under the European Union's Horizon 2020 research and innovation program (882673).

## Author contributions

L.-H.C., S.G., and D.N. conceived the study, designed the experiments, and wrote the manuscript. L.-H.C. developed the Nano-C assay and performed most experiments. S.G. developed the dedicated Nano-C and ChIP-seq analysis pipelines and analyzed most of the data. A.P. and D.H. adapted and performed the RCL polymer simulations and contributed the section on RCL modeling. J.M.L. and E.F.J. performed Oligopaint FISH experiments and provided input on the manuscript. M.M., J.E., and N.L. created and analyzed mESC lines with CBS deletions. M.P. and V.P. performed and analyzed additional ChIP-seq experiments. J.D. and A.L. performed and analyzed Cryo-EM experiments. S.B. provided essential input for the development of Nano-C and provided input on the manuscript.

## Competing interests

The authors declare no competing interests.
