## [Peer Review File · Nature Communications]

Multi-feature clustering of CTCF binding creates robustness for loop extrusion blocking and Topologically Associating Domain boundariesEditorial Note: This manuscript has been previously reviewed at another journal that is not operating a transparent peer review scheme. This document only contains reviewer comments and rebuttal letters for versions considered at *Nature Communications*.

REVIEWERS' COMMENTS

Reviewer #1 (Remarks to the Author):

The manuscript has much improved and I now support publication.

Reviewer #2 (Remarks to the Author):

The revised manuscript is toned down and largely improved. However, most ChIP-seq or ChIP-nexus peaks may not contain multiple CBS elements. Without convincing ChIP-nexus or ChIP-exo data to show multiple subpeaks within a single ChIP-seq peak, it is impossible to show a single peak contains multiple CBS elements (Figures 1 and 2; Extended Data Figure 1).

Reviewer #3 (Remarks to the Author):

The authors have addressed most of my comments and I think that the revised manuscript is strongly improved.

I have a few remaining minor comments:

1. In the abstract and introduction, the authors talk about "enrichment of three features of CTCF binding" without specifying explicitly what these features are. It takes until lines 181-183 before these features are clearly described. I think it would be more convenient for the readers if this information (which is one of the main contributions of the paper) is specified earlier.
2. Extended Data Figure 4B. The comparison to other multi-way 3C methods does not seem quite right. I do not think that Tri-C experiments require 14 NextSeq runs to generate ~200M reads. My understanding is that Tri-C works similar to Capture-C, and that it supports multiplexing of samples (in addition to multiplexing of viewpoints) with different indices in a single NextSeq run (which then results in multiple independent fastq files).

We thank the three reviewers for their renewed efforts to evaluate our strongly revised manuscript. We are happy to hear that they all agree that the manuscript has much improved.

We have provided detailed comments below each of the specific comments.

Reviewer #1:

Remarks to the Author:

The manuscript has much improved and I now support publication.

Reviewer #2:

Remarks to the Author:

The revised manuscript is toned down and largely improved. However, most ChIP-seq or ChIP-nexus peaks may not contain multiple CBS elements. Without convincing ChIP-nexus or ChIP-exo data to show multiple subpeaks within a single ChIP-seq peak, it is impossible to show a single peak contains multiple CBS elements (Figures 1 and 2; Extended Data Figure 1).

We are of the opinion that our reanalysis of the CTCF SLIM-ChIP data in mESCs, obtained with a technology that is very similar to ChIP-nexus and ChIP-exo, does convincingly show that many CBS exist within the genome where multiple motifs can be bound. Particularly, we don't see alternative explanations for the observations that:

- 1. motifs that are present together within the same CBS can have similar signal strength (or even higher) as compared to the single motif category (fig. 2c, with motifs grouped on their MEME significance score and read counts normalized to motif number in each category).*
- 2. overlapping motifs can create a reduction in the number of basepairs that are protected (fig. 2d).*

Although ChIP-exo is not generally interpreted at the single-peak level, we have provided below the SLIM-ChIP signal over CBS 20326 (below). Here we can observe peaks of protection throughout a large part of the CBS. This includes peaks on the forward strand that localize downstream of the most significant motif 2 and peaks on the reverse strand that localize far away from the same motif. We interpret this result as a direct confirmation for CTCF binding at motifs 2, 3, 4 and 5. This outcome fits with both our ChIP-seq in WT cells and the outcome of our genome editing and ChIP-qPCR experiments on the same CBS (figure below and Fig. 2a).

CTCF ChIP-seq and SLIM-ChIP - CBS 20326 (single basepair resolution)

Reviewer #3:

Remarks to the Author:

The authors have addressed most of my comments and I think that the revised manuscript is strongly improved.

I have a few remaining minor comments:

1. In the abstract and introduction, the authors talk about "enrichment of three features of CTCF binding" without specifying explicitly what these features are. It takes until lines 181-183 before these features are clearly described. I think it would be more convenient for the readers if this information (which is one of the main contributions of the paper) is specified earlier.

The reviewer makes a very good point. We now explicitly mention the three features of CTCF binding in both the abstract and at the end of the introduction.

2. Extended Data Figure 4B. The comparison to other multi-way 3C methods does not seem quite right. I do not think that Tri-C experiments require 14 NextSeq runs to generate ~200M reads. My understanding is that Tri-C works similar to Capture-C, and that it supports multiplexing of samples (in addition to multiplexing of viewpoints) with different indices in a single NextSeq run (which then results in multiple independent fastq files).

We thank the reviewer for pointing out this oversight. When revisiting our data, we realized that individual GEO accession numbers consisted of around 20M reads each. This indeed indicates that multiple replicates were pooled within a single Next-seq run.

We have changed the Supplementary figure 4B to mention replicates, rather than runs.

While looking into this question, we noticed an error in our analysis that increased the number of multi-contact reads (but not 1-way reads) that were reported for MC-4C and Tri-C. We have adjusted these numbers in Supplementary figures 4B and 4D.